# Light-triggered multi-joint microactuator fabricated by two-in-one femtosecond laser writing

Chen Xin[1,2,4], Zhongguo Ren[1,4], Leran Zhang[1], Liang Yang[3], Dawei Wang[1], Yanlei Hu[1], Jiawen Li [1], Jiaru Chu[1], Li Zhang[2] & Dong Wu [1] ✉

Inspired by the flexible joints of humans, actuators containing soft joints have been developed for various applications, including soft grippers, artificial muscles, and wearable devices. However, integrating multiple microjoints into soft robots at the micrometer scale to achieve multi-deformation modalities remains challenging. Here, we propose a two-in-one femtosecond laser writing strategy to fabricate microjoints composed of hydrogel and metal nanoparticles, and develop multi-joint microactuators with multi-deformation modalities (>10), requiring short response time (30 ms) and low actuation power (<10 mW) to achieve deformation. Besides, independent joint deformation control and linkage of multi-joint deformation, including co-planar and spatial linkage, enables the microactuator to reconstruct a variety of complex human-like modalities. Finally, as a proof of concept, the collection of multiple microcargos at different locations is achieved by a double-joint micro robotic arm. Our microactuators with multiple modalities will bring many potential application opportunities in microcargo collection, microfluid operation, and cell manipulation.

Flexible joints exist widely in natural organisms, and the cooperation between joints leads to elaborate movements and the realization of variable functions[1,2]. As a typical representative, humans can use multi-joint limbs to complete walking, running, and jumping actions, and multi-joint fingers to manipulate various objects[3]. The programmable conformations of flexible multi-joint have inspired researchers to develop a variety of soft components for broad applications in soft robots[4–11], wearable devices[12–15], and biomedical devices[16,17]. For example, terrestrial robots with multiple shape memory alloy joints can realize linear/curvilinear crawling, walking, turning, and jumping by laser-inducing[18]. Flexible hands with multi-joint are developed to help disabled people perform manipulations on different target objects[19]. As actuators and devices get miniaturized due to the requirements of

integration and intelligence, responsive and controllable joints with micrometer size are brought to the forefront. However, due to the lack of suitable processing methods and smart materials, integrating multi-joints into actuators at the microscale for manipulating microobjects remains challenging.

To date, hydrogels exhibit high biocompatibility, strong environmental adaptability, and mechanical properties comparable to biological tissues[20–22], making them advanced in biomedical applications[23,24]. More importantly, hydrogels with stimuli-responsive deformability have emerged as highly competitive materials for microactuator fabrication[25–27]. The triggering modes of responsive hydrogels are currently classified as chemical[28–33], thermal[34–37], and light-triggering modes[38–41]. Chemical and thermal-triggered hydrogel can achieve

[1]Key Laboratory of Precision Scientific Instrumentation of Anhui Higher Education Institutes, CAS Key Laboratory of Mechanical Behavior and Design of Materials, Department of Precision Machinery and Precision Instrumentation, University of Science and Technology of China, Hefei 230026, China. [2]Department of Mechanical and Automation Engineering, The Chinese University of Hong Kong, Hong Kong 999077, China. [3]Suzhou Institute for Advanced Research, University of Science and Technology of China, Minde Building, Renai Road, 215123 Suzhou, P. R. China. [4]These authors contributed equally: Chen Xin, Zhongguo Ren. ✉e-mail: dongwu@ustc.edu.cn

reversible shape transitions, such as screwing and untwisting of a helix[40], as well as closing and opening of a gripper[42,43]. However, chemical and thermal triggering modes tune the whole environment in which the hydrogel is placed, which ultimately leads to indiscriminate deformation of the entire hydrogel microstructures and the incapability of independent control of individual parts[44,45]. In addition, the relatively slow diffusion of heat and chemical components results in the hydrogels exhibiting long response time[25,46]. In contrast, the light-triggered hydrogels exhibit precise local stimulation and short response time[47–49]. Recently, light-triggering as a flexible and remote-control mode is widely used to drive soft actuators[50,51]. Advanced 2D structured light and dynamic light scanning methods promote the miniatured actuators to increase their deformation modalities[18,52] (Supplementary Table 1). However, precisely tuning the light distribution to independently stimulate the local region deformation of the 3D microactuator (<100 μm) for achieving multi-modality in 3D space remains challenging. Building multiple joints in microactuators and achieving independent control of each joint is an attractive way to increase the microactuator's modalities and manipulation capabilities. For example, a multi-joint robotic arm in a mechanical factory can achieve multi-dimensional deformation and operate multiple objects at different locations by control of each joint. Thus, constructing multiple microjoints and achieving their independent deformation control are crucial and promising for the application development of functional microactuators.

In this work, we propose a two-in-one femtosecond laser processing strategy to construct light-triggered multi-joint microactuators (MJMAs) with multiple deformation modalities (>10). The two-in-one processing strategy includes the construction of bio-inspired microjoints using asymmetric two-photon polymerization techniques[53–56] and the deposition of silver nanoparticles (Ag NPs) onto hydrogel joints based on photoreduction techniques. Due to the strong photothermal conversion effect of the densely stacked Ag NPs, the microactuator exhibits short response time (30 ms) and low actuation laser power (<10 mW), even lower than the processing laser power (33 mW) of the microactuator. In addition, the independent deformation of each microjoint could be achieved by precise multi-foci light distribution control. With the cooperation of multiple joints in 3D space, the humanoid microactuator demonstrates multiple deformations modalities, such as raising arms, legs, and feet. Finally, as a proof of concept, a double-joint micro robotic arm (MRA) can be used to manipulate multiple microcargos at different locations, similar to a robotic arm in a factory. We believe that these light-triggered MJMAs will provide general technical tools for the fields of microcargos manipulation, microfluidic chips, and microoptics.

## Results

### Design and Fabrication of light-triggered multi-joint microactuators by two-in-one laser printing

The flexible joint deformation of humans enables us to complete various modalities, such as walking, running, and jumping. Inspired by these joint bending behaviors (Fig. 1a), a photothermal-responsive hydrogel is chosen to construct flexible joints. A two-in-one strategy, consisting of precise femtosecond (Fs) laser printing of hydrogel and selective laser reduction of metal nanoparticles is proposed to manufacture reconfigurable MJMAs. Figure 1b shows the multi-deformation modalities of a humanoid MJMA. The light-triggering method enables each joint to be stimulated independently, as well as linkages of multiple joints, facilitating the microactuator with multiple human-like modalities. The fabrication of a humanoid MJMA is composed of two steps. In the first step, a 3D hydrogel-based humanoid microactuator is fabricated by an asymmetrical TPP process (Fig. 1c and Supplementary Fig. 1). Specifically, the hydrogel joints consist of both low and high cross-linking density parts with large and small shrinking ratio separately. In the second step, Ag NPs are selectively deposited on the surface of the humanoid robot joint by light

reduction (Fig. 1d and Supplementary Fig. 2). In this step, the hydrogel structure is immersed in an Ag ink, and silver ammonium ions are reduced to Ag NPs by absorbing photons, which are uniformly attached to the hydrogel joints with average roughness of 58.5 nm (Supplementary Fig. 3). SEM images before and after the reduction of Ag NPs on the joint surface of the humanoid robot are presented, where the contours of Ag NPs are marked in yellow (Fig. 1d). Energy dispersive spectrometer (EDS) images also demonstrate that the joint is mainly composed of C and Ag elements, where the pattern of the Ag NPs is controllable (Supplementary Fig. 4). In this way, the asymmetric hydrogel constitutes the joint deformation unit, and the metal nanoparticles constitute the light-to-heat conversion unit. It is worth mentioning that both hydrogel polymerization and Ag NPs reduction are realized in the same Fs laser system without any additional modifications, so we call it a two-in-one manufacturing strategy.

To manipulate MJMAs, a precise 3D light distribution control technique is essential for the independent stimulation of multiple joints. Liquid crystal spatial light modulators (Lcos-SLM) with light phase and polarization modulation are widely used to construct different spatial beams[57–61]. Therefore, multi-foci beams are modulated using the spatial light modulation technique to precisely control the deformation of each joint (Fig. 1e), which are generated by the SLM loaded with a series of prepared computer-generated holograms (CGHs). The measured multi-foci spatial position and intensity are consistent with the simulated results, where the diameter of each light foci is ~2.3 μm (Supplementary Fig. 5). In the composite joint structure, the thermo-responsive hydrogel is a smart material that can exhibit contraction in response to heat, and the Ag NPs function as the photoreceptors and photothermal agents, which convert light energy to thermal energy to heat the thermo-responsive hydrogel. When the irradiated joint temperature surpasses the low critical solution temperature (LCST) of the hydrogel, the hydrogels expose the hydrophobic groups extruding the inside water molecules to produce a strong contraction[62]. Since the joint consists of a low cross-linking layer and a high cross-linking layer, the low cross-linking layer has a stronger contraction ability. Therefore, the hydrogel joint generates asymmetric volume contraction force resulting in bending toward the low cross-linking side (Supplementary Fig. 6). To illustrate the programmable configurations of MJMA, an MJMA with four microjoints is manipulated dynamically over time by a series of multi-focal beams (from one to four foci). When different numbers and spatial distributions of the multi-focal beam are adjusted, the humanoid MJMA can display various subsequent modalities (Supplementary Movie 1).

### Controllable deformation of hydrogel microactuator by precise light-triggering

In this study, the hydrogel is mainly composed of a thermal responsive monomer N-Isopropyl acrylamide (NIPAM)[62,63], a crosslinking agent Methylene-Bis-Acrylamide (MBA), and an initiator Diphenyl (2,4,6-trimethyl benzoyl) phosphine oxide (TPO). Due to the poor mechanical strength of printed NIPAM[64], polyvinylpyrrolidone (PVP, K30) is added to improve the mechanical ability of the hydrogel by constructing hydrogen bonds with PNIPAM. The PVP significantly enhances the mechanical properties of hydrogel microstructures (Supplementary Fig. 7) while having essentially no negative impact on the thermosensitivity and photothermal sensitivity of the hydrogel (Supplementary Fig. 8). In addition, the addition of PVP significantly increases the viscosity of the hydrogel precursor (0.073 Pa·s) to prevent low processing quality due to the flow of the hydrogel during processing (Supplementary Fig. 9). To investigate the contraction capability of the hydrogel upon the triggering of heat and light, circular plates with a diameter of 20 μm and a thickness of 3 μm are fabricated. Notably, these plates are supported by cylinders with a diameter of 4 μm and a height of 10 μm to avoid interference from the substrate regarding its expansion and contraction. A significant contraction can be observed

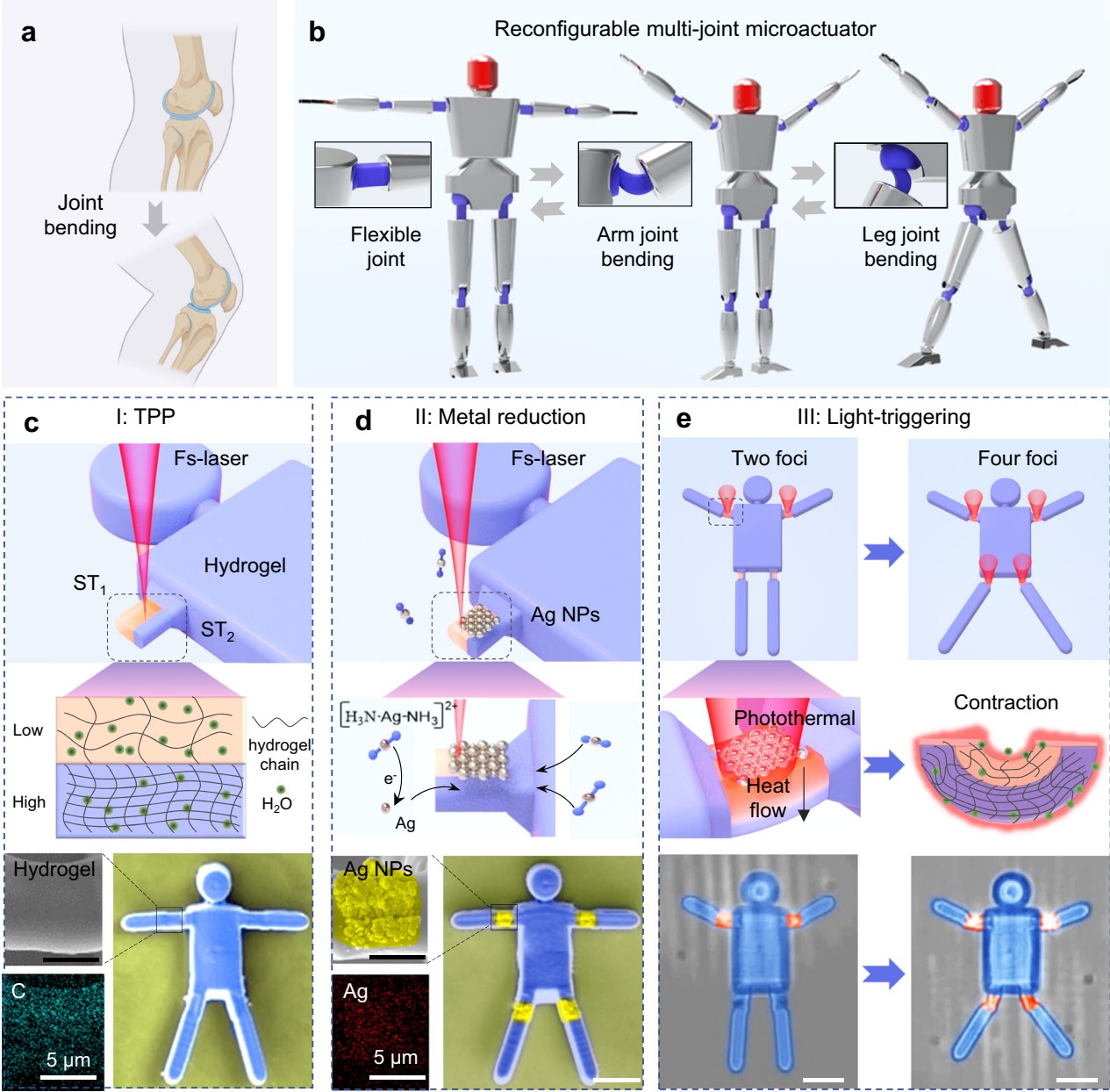

**Fig. 1 | Design and Fabrication of light-triggered multi-joint microactuators (MJMAs) by two-in-one laser printing. a** The human arm relies on joint bending to achieve hand raising. **b** The schematic diagram of a light-triggered humanoid MJMA showing multiple modalities. **c** Laser printing responsive hydrogel to construct the MJMA skeleton. The joint is composed of two parts with different cross-link densities, where orange and blue represent low and high cross-link densities, respectively. $ST_1 = 1\,ms$ and $ST_2 = 3\,ms$ are single-point scanning time during processing. **d** Laser reduction of Ag NPs for photothermal conversion, in which silver ammonium ions absorb photons and are reduced to Ag NPs. SEM images show the

corresponding materials of hydrogel MJMAs (blue) and Ag NPs (yellow), respectively. **e** Two foci and four foci are modulated by the Gerchberg-Saxton algorithm to control the deformation of two and four joints, where the spatial position and intensity of the focal spot can be flexibly adjusted. When Ag NPs are irradiated by NIR light, a large amount of heat transfer from Ag NPs to hydrogel will be generated. The parts of the hydrogel with low cross-link density contract more than the part of the hydrogel with high cross-link density, thus causing directional bending of the hydrogel joint. All scale bars are 20 μm.

when the microplates are triggered by heated liquid and NIR light (Fig. 2a). A shrinking ratio ($\varepsilon$) is defined to quantitatively evaluate the deformation capability of hydrogel microplates as:

$$\varepsilon = \frac{D_1 - D_0}{D_1} \tag{1}$$

where $D_0$ and $D_1$ are the diameter of shrunken and original microplates, respectively. For a better understanding of the temperature distribution and reasonable design of MJMA, we develop a Multiphysics model that takes large deformation, mass diffusion, and heat transfer

into account to numerically simulate the deformation of the microplate, which could provide a reliable prediction of the subsequent deformation of the microstructure. Numerical calculations show that the microplate shrinks with increasing temperature through heat transfer (Fig. 2b). Subsequently, to investigate the shrinkage property of hydrogel, we explore the effect of direct laser writing (DLW) parameters on hydrogel shrinkage including processing laser power (PLP), scanning point step (SPS), and scanning time (ST). DLW is a point-by-point scanning process so that the 3D microstructure is built up from one focal spot to another[65,66]. In this way, PLP is the power of the focal spot, SPS is the distance between two adjacent focal spots, and ST is the

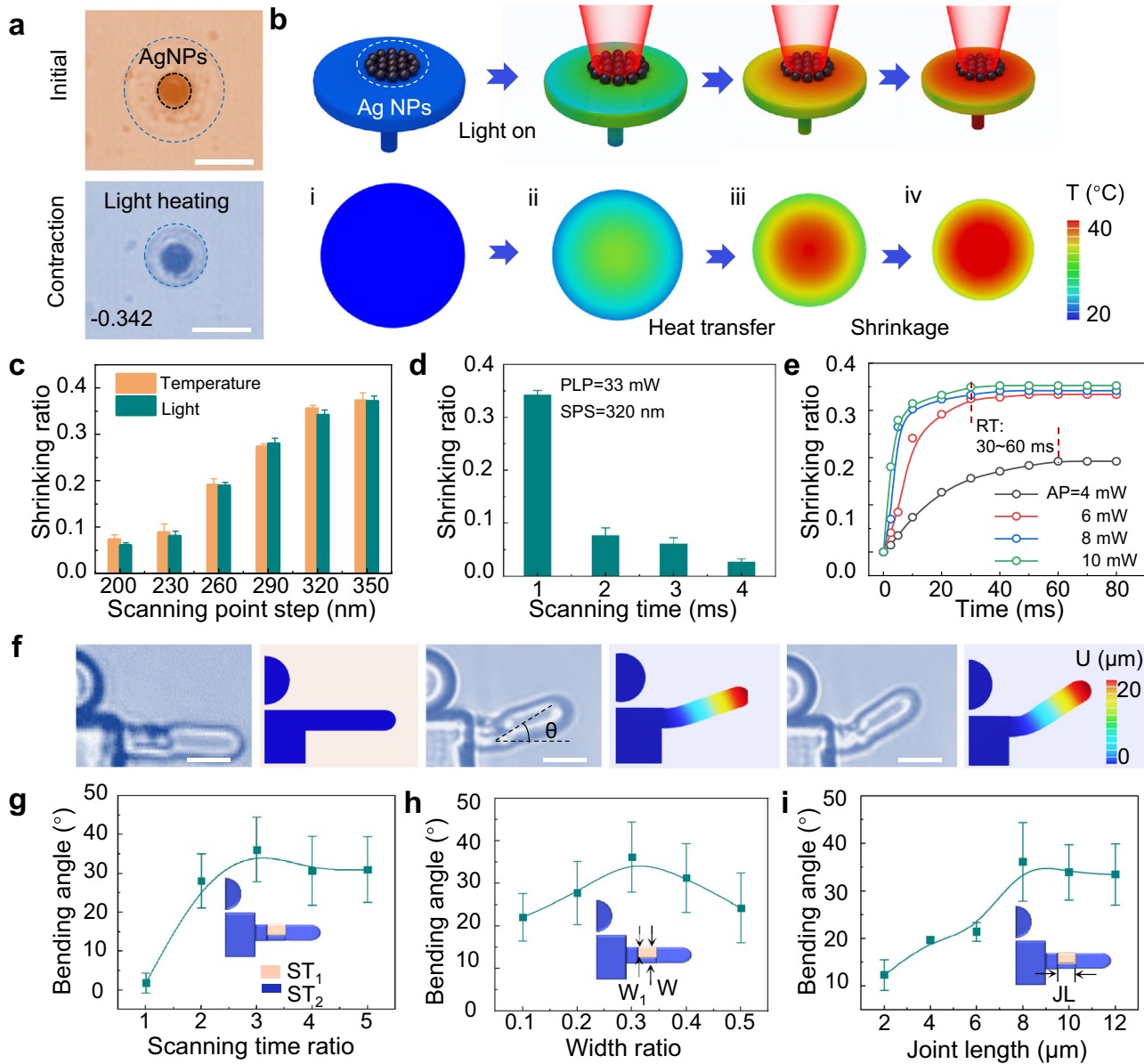

**Fig. 2 | Controllable deformation of hydrogel microactuator by precise light heating. a** Optical microscopy images of hydrogel microplate contracting under light heating. **b** Schematic and simulation results of Ag NPs heating by light absorption and hydrogel deformation by heating transfer. **c** The shrinking ratio of hydrogels under local laser heating is consistent with global liquid heating. PLP and SPS represent processing laser power (PLP) and scanning point step (SPS), respectively. **d** As the single-point scanning time increases from 1 ms to 4 ms, the shrinkage of the hydrogel decreases from 0.34 to 0.03. RT represents the response time of the microplate. **e** As the local actuation laser power increases to 10 mW, the

hydrogel demonstrates a short response time of 30 ms. **f** Time-lapse images of the arm joint bending angle ($\theta$) increasing with the light-triggering time. The quantitative relationship between joint bending angle and scanning time ratio (STR) (**g**), width ratio (WR) (**h**), and joint length (JL) (**i**), when STR = 3, WR = 0.3, and JL = 8 μm, the joint reaches the maximum bending angle. $ST_1$ and $ST_2$ are single-point scanning time to process different part of the microjoint. $W_1$ and $W$ are the width of the low crosslink density portion and the whole microjoint, respectively. All scale bars, 10 μm. Error bars stand for the standard error ($n = 3$).

dwell time of the focal spot at each position. To balance the mechanical strength and shrinkage capability of the microstructure, the processing parameters PLP = 33 mW and SPS = 320 nm are chosen for processing microstructures (Supplementary Fig. 10). In our study, we find that for microplates processed with different SPS, their shrinking ratio induced by light is comparable to that triggered by liquid temperature (Fig. 2c). In addition, ST as a changeable parameter could control the shrinking rate and mechanical properties of the microstructure. When the ST increases from 1 ms to 3 ms, the hydrogel shrinking ratio decreases from 0.34 to 0.06, while Young's modulus increases from 1.55 Mpa to 6.34 Mpa. (Fig. 2d and Supplementary Fig. 11). The reason for the decrease in the shrinking ratio is that as the ST increases, the

crosslinking density of the hydrogel increases, resulting in a higher porosity of the hydrogel (Supplementary Fig. 12). Since it is easy and convenient to adjust ST during printing, we choose it as the final parameter to flexibly control the shrinking ratio of the microplate. In addition to the processing parameters of the hydrogel, the thickness of the Ag NPs layer produced by laser reduction has an important effect on the shrinking ratio of light-triggered microstructure. When the layer thickness of Ag NPs is <200 nm, the shrinking ratio of the microplate is 0.29. When the layer thickness of Ag NPs is >300 nm, the shrinking ratio of hydrogel reaches the maximum value (Supplementary Fig. 13). Since Ag NPs generate a strong electric field enhancement and thus a large amount of heat when exposed to light radiation

(Supplementary Fig. 14), the joints present short response time and low actuation power (AP). When the AP increases from 4 mW to 10 mW, the response time of microplates deformation decreases from 60 ms to 30 ms (Fig. 2e, Supplementary Fig. 15, and Supplementary Movie 2). Compared to the temperature-triggering mode that depends on the heating and cooling of the surrounding liquid[67], the local light-triggering could rapidly switch the temperature of local regions surrounding the microstructures, resulting in the microstructure exhibiting shorter deformation time. To investigate the reversibility of the actuation process, we conduct more than 100 cycles of switching on/off the illumination on the sample, and no obvious fatigue in the optical images is recorded after each stimulation, indicating high stability. In addition, according to the optical and SEM images, no Ag NPs are dislodged during multiple actuations (Supplementary Fig. 16 and Supplementary Movie 3). Therefore, we leverage the light-triggering mode to remotely actuate the soft MJMAs below.

Apart from isotropic hydrogel contraction, the anisotropic contraction of microstructures can be flexibly controlled. Here, a joint of a humanoid MJMA is chosen to investigate anisotropic bending (Fig. 2f and Supplementary Movie 4). The joint is divided into two parts with different crosslinking densities and anisotropic porosity (Supplementary Fig. 17). The bending degree of the joint is mainly related to three factors, including scan time ratio (STR), width ratio (WR), and joint length (JL). As shown in Fig. 2g, the joint deformation reaches a maximum value of ~35° when STR is 3 (Supplementary Fig. 18). In addition, the joint WR also affects the bending degree of the joints. With the continuous increase of WR, the bending angle of the joint first increases and then decreases (Fig. 2h and Supplementary Fig. 19). Finally, the joint bending degree is positively related to the JL, however, when the joint reaches the maximum length of the light-triggering, the degree of deformation keeps stable (Fig. 2i and Supplementary Fig. 20). Apart from horizontal deformation, we also realize the multiple vertical deformations of water strider-like MJMAs (Supplementary Fig. 21 and Supplementary Movie 5). Here, the MJMA is composed of four joints of upper and lower layers with different cross-linking densities and a homogeneous body. The hybrid water strider-like MJMA exhibits a series of configurations under dynamic illumination stimuli, which agrees well with the simulation results. In this process, the deformation of the four joints maintains a good consistency, and the deformation angle is stable at ~36°.

## Multi-joint linkage deformation of a humanoid multi-joint microactuator

To further prove that this simple method can be applied to more complex deformation control, a humanoid soft MJMA with eight joints is demonstrated. By controlling the gray value of CGH, the intensity of actuation light foci can be flexibly controlled, so that different joints have different degrees of deformation. For example, one joint on the left arm close to the body is stimulated with 100% light power. Meanwhile, the other symmetrical joint is stimulated with 20%, 40%, 60%, 80%, and 100% light power (Fig. 3a, b). As a result, the left arm always maintains the maximum bending angle of ~31°, and the right arm bending angle monotonously increases with the increased excitation light intensity. Notably, when the joint on the right arm close to the body is excited with 100% light intensity, the right arm bending deformation saturates and also reaches a maximum value of ~31° (Fig. 3c). By increasing the number of actuation light foci, we can accomplish more complex modality reconfigurations of the humanoid soft MJMA with two joints on each limb. We modulate the actuation light foci pattern to stimulate different joint numbers, including one, two, and four foci patterns (Supplementary Fig. 22). For different numbers of foci, we need to load grayscale weights on the hologram to keep the radiation energy consistent (Supplementary Fig. 23). Here, we define the bending angles of the joints close to and far from the body as θ1 and θ2, respectively. As an example, we first modulate the single

foci mode incident light onto the joint on the right arm close to the body to make the right arm bend to form modality i (Fig. 3d), and then the configuration of modality ii is accomplished in situ by switching actuation light foci onto the joint on the right arm far from the body. In this way, multiple deformation modalities can be obtained in the same humanoid soft MJMA under the one, two, four, and eight foci stimuli (Fig. 3e, f). As expected, as more joints are triggered, the total deformation done by the actuator's execution end increases, from 28° for joints number (JN) = 1 to 45° for JN = 2 (Fig. 3g, h). More importantly, the bending angle of four joints on the left and right sides of the humanoid MJMA maintains a good consistency, and the error does not exceed 1° (Fig. 3h and Supplementary Movie 6). These results suggest that noninvasive and remote reprogramming can be used to tune the modality of soft MJMA. To provide the state-of-the-art of MJMAs, soft actuators have been comprehensively summarized in Supplementary Table 1. Compared to the existing soft actuators, our microactuators constructed by two-in-one Fs printing demonstrate multiple modality numbers (>10), much smaller 3D structure sizes (<100 μm), precise local region response (30 ms), and low actuation power (<10 mW).

## Multi-joint linkage deformation in 3D space

In addition to the deformation in a single plane, the proposed method also allows the orderly programmed deformation of the soft MJMA in 3D space. To demonstrate the feasibility of programming the 3D architectures, a distinct humanoid soft MJMA is fabricated, where the joints ($J_3$, $J_4$, $J_7$, $J_8$) on the arms close to the body and the legs far from the body consists of horizontally asymmetric crosslink density layers, while the joints ($J_1$, $J_2$, $J_5$, $J_6$) on the arms far from the body and the legs close to the body consist of vertically asymmetric crosslink density layers (Fig. 4a). With multi-focal illumination, the humanoid MJMA could link horizontal and vertical deformations of multiple joints. It is worth noting that the multi-foci here are generated in 3D space. For example, the double foci hologram for the vertical deformation of the leg and horizontal deformation of the feet, are composed of two parts. One focal point (point 1) of the bifocal is focused on the $Z = 0$ μm plane, and the other focal point (point 2) is focused on the $Z = 15$ μm plane (Fig. 4b). Through the light intensity simulation, the light density information on different XY planes is extracted. The two points are normally distributed in their respective planes, and there will be no interference between the two points (Fig. 4c and Supplementary Fig. 24). Thus, if the same MJMA is stimulated with relevant multi-focal illumination by switching 3D multi-foci CGHs, it is expected to present deformations in different planes. The constructed finite element simulation is used to predict the experimental results, which enables us to design the 3D deformation of MJMAs (Fig. 4d). Take modality ii (Fig. 4e) as an example, the joints on the leg close to the body are illuminated to cause the leg to raise upwards in the vertical plane, and the joints on the leg far from the body are excited to further induce the front end of the same leg to bend in the horizontal plane (Supplementary Movie 7). Other 3D deformations can be generated by similar designs to cope with different application requirements.

## A light-triggered multi-joint micro robotic arm

In recent years, with the development of science and technology, robot arms have become an important device to assist people in production. They can perform various operations on objects on assembly lines in factories, such as picking, transferring, assembling, and unloading. Although the traditional rigid robotic arm shows strong manipulation capability, building a soft robotic arm at the micro-scale and completing the operations of multiple microcargos remain challenging. Herein, a micro robotic arm (MRA) with double joints is developed to manipulate microcargos located at different positions (Fig. 5a). When the joint far away from the base is excited, the front half of the MRA momentarily bends toward the side with lower cross-linking density, and the bending angle (θ1) of the free end is

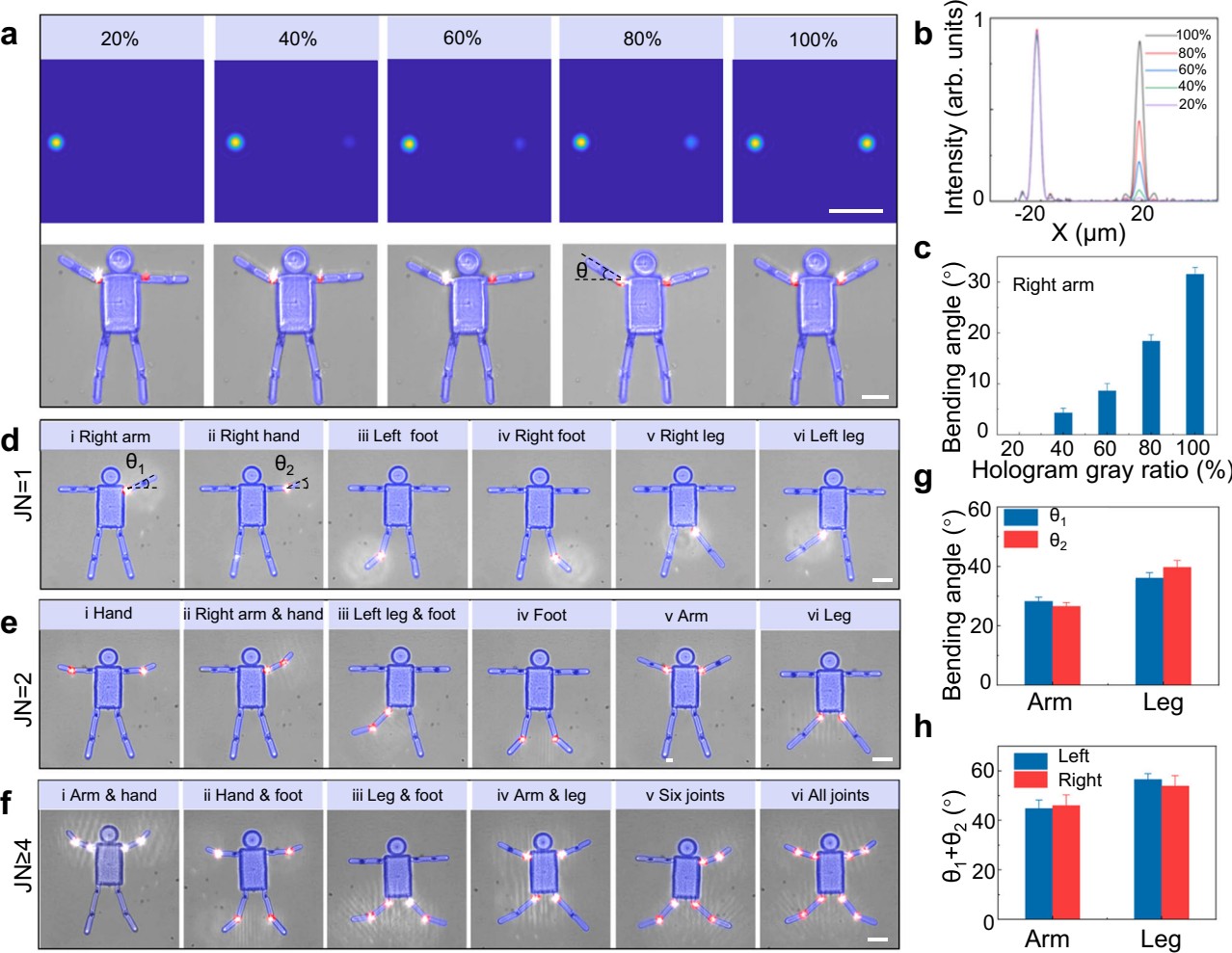

**Fig. 3 | Multi-joint linkage deformation of a humanoid multi-joint micro-actuator (MJMA). a** Simulation images of the two foci intensity from 20% to 100% and the corresponding microscope images of the bending degree of the two arms of the MJMA. θ represents the bending angle of the microjoint. **b** By controlling the grayscale of the focal hologram, two foci with different intensities are obtained. **c** As the grayscale of the hologram increases from 40% to 100%, the bending angle of the MJMA right arm increases from 4° to 31°. **d** Single-joint, **e** double-joint, and **f** multi-joint deformation modalities of humanoid MJMAs. The bending angles of the joints close to and far from the body are defined as θ1 and θ2. JN is denoted as joints number. **g, h** As the number of deformed joints increases, the total bending angle of the MJMA increases. In multiple multi-joint morphing modalities, the left and right sides of the MJMA exhibit high morphing consistency. JN represents the joint number of the microactuator. Scale bars, 20 μm. Error bars stand for the standard error (n = 3).

measured to be 28° (Fig. 5b, c, modality i). If the CGH is switched, the joints close to the base are stimulated, causing the same MRA to be reprogrammed to another deformation modality, with a bending angle (θ2) up to 32° (Fig. 5b, c, modality ii). Eventually, the corresponding double foci CGH is generated by the Gerchberg-Saxton (GS) algorithm, and the sum of the bending angle of both joints are almost 45° (Fig. 5c, modality iii). Subsequently, dynamic multi-foci structured light fields modulate the reconfigurations of soft MRAs to enable the collection of multiple microobjects at different locations, the gripping force originates from the adhesion of the thermo-responsive hydrogel. As an example, we fabricate a soft MRA and two microparticle bases (side length 6 μm). The function of the microparticle base is to support the targeted cargo to the height corresponding to the robotic hand, and its location is determined by the length, bending angle, and bending direction of the MRA. The schematic illustration of the collection procedures is shown in Fig. 5d, including the initial state, the first-position collection, the second-position collection, and the final state. The snapshots of different procedures are shown in Fig. 5e. In the initial state, the magnetic microparticles fall on the microparticle base with the help of an external magnetic field. When the joint close to the base is illuminated, the MRA drives the robotic hand to approach

the first microparticle base, triggering the first-position microparticle collection. The microparticle is transported to the original position in real time once the illumination is turned off. To implement the second-position microparticle collection, the joint far from the base is excited, and thus, the MRA is compelled to bend in the opposite direction to gather the second-position microparticle. When the excitation is removed, the microparticles at different locations are collected together. Furthermore, we can also realize the collection of targeted microparticles on the same side but at different locations (Fig. 5f, g and Supplementary Movie 8). The reprogrammable MRA that can controllably accomplish the collection of microparticles at different locations in situ is promising for miscellaneous applications in precision assembly, cell operation, and cargo collection.

## Discussion
Although many efforts have been devoted to developing light-triggered soft actuators[42,68–70], the open challenges of limited 2D morphologies, slow response speed, and inaccurate light-triggering have restricted their further innovation. The MJMAs in the current work demonstrate multi-deformation modalities (>10), low actuation power (<10 mW), and short response time (30 ms) in 3D space and

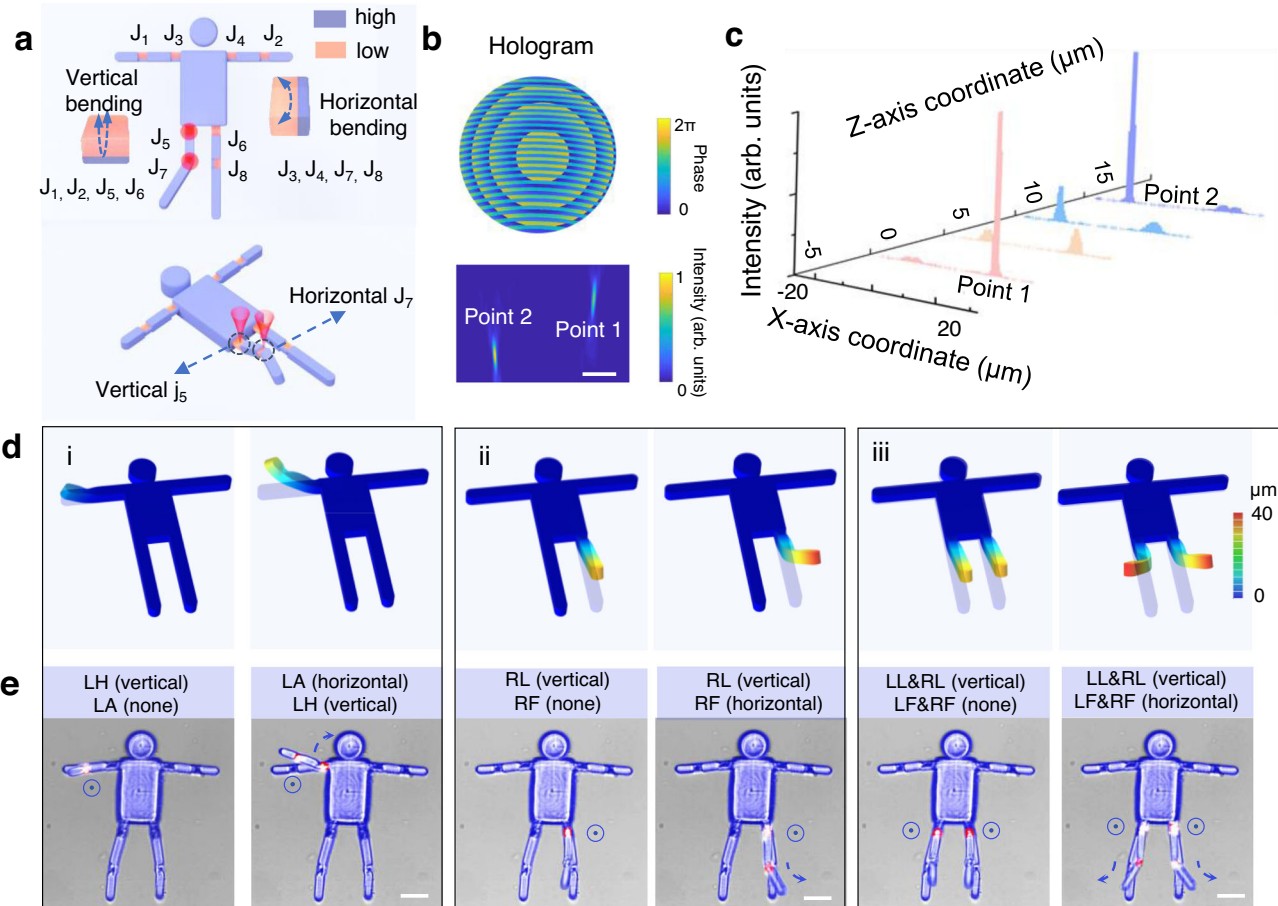

**Fig. 4 | Multi-joint linkage deformation in 3D space. a** The two joints of the legs of the humanoid multi-joint microactuator (MJMA) are driven in different planes to complete the deformation in the vertical and horizontal directions respectively. $J_1$, $J_2$, $J_3$, $J_4$, $J_5$, $J_6$, $J_7$, $J_8$ are the joints on the arms and the legs. **b** Double foci can be obtained on different Z planes by superimposing spherical wave phases at partial focal points. **c** Double foci distribution on different planes, point 1 is focused on the plane of Z = 0 μm, and point 2 is focused on the plane of Z = 15 μm. **d** Finite element simulation results of double-joint linkage deformation of a humanoid MJMA in 3D space. **e** Linkage deformation of two joints driven by two foci in different XY planes. LH, LA, RF, RL, LL, LF represent left hand, left arm, right foot, right leg, left leg, and left foot, respectively. Scale bars, **b** 10 μm, **e** 20 μm.

manipulate multiple microcargos at different locations. In particular, by enabling multi-foci beams through SLM, hydrogel microstructures allow for highly complex soft robotics manipulations at the micro-scale. These structures demonstrate strong deformable capabilities, exceeding the ability of multi-beam optical tweezers to manipulate microcargos. In addition, hydrogel microactuators could also be adapted to the needs of manipulation in dynamic environments by taking advantage of the locomotion strategies that exist in nature, such as sperm and protozoa. For example, we could introduce magnetic materials into existing microactuators to build movable microrobot systems with multiple programmable modalities, that is, magnetic fields control locomotion, and light fields control deformation.

In summary, we propose a two-in-one processing strategy to construct light-triggered MJMAs, which exhibit multiple deformation modalities (>10) in 3D space. The strong photothermal conversion effect of densely stacked Ag NPs results in low actuation power (10 mW) and short response time (30 ms) of the MJMAs. In addition, we not only realize the independent deformation control of each joint of MJMAs, but also can complete multi-joint linkage control in 3D space, thus making MJMAs exhibit multiple deformation modalities, including raising hand, leg, and foot. Ultimately, as a proof-of-concept, the double-joint micro robotic arm can collect microparticles at different locations. Certainly, the light-triggering mode is accurate and pollution-free, however, its limited penetration is the main limitation of its application in non-transparent environments. Therefore, the use of

magnetic or ultrasonic heat to actuate MJMAs may also be a good choice[71,72]. In the future, we believe that constructing intelligent MJMAs would have the potential to enable advanced applications in a wide variety of medical, science, and engineering fields.

## Methods

### Materials
N-isopropylacrylamide, N, N′-methylenebis (acrylamide), Diphenyl (2,4,6-trimethyl benzoyl) phosphine oxide, ethylene glycol, Poly-vinylpyrrolidone K30, Rhodamine 6 G, trisodium citrate, aqueous ammonia, and silver nitrate were purchased from Aladdin. The other materials were purchased from China National Pharmaceutical Group Corporation.

### Preparation of silver precursor
Firstly, 14.1 mg of silver nitrate aqueous and 13.4 mg of trisodium citrate were added and stirred in 1 mL of deionized water. The silver precursor was subsequently prepared by dripping 100 μL of aqueous ammonia (25–28% NH₃ in H₂O) into a mixture under stirring until a clear solution was formed.

### Preparation of pNIPAM precursor
The precursor was prepared by dissolving 400 mg N-isopropyl acryla-mide, 30 mg N, N′-methylenebis (acrylamide), 30 mg Diphenyl(2,4,6-trimethylbenzoyl) phosphine oxide, and 50 mg of Polyvinylpyrrolidone

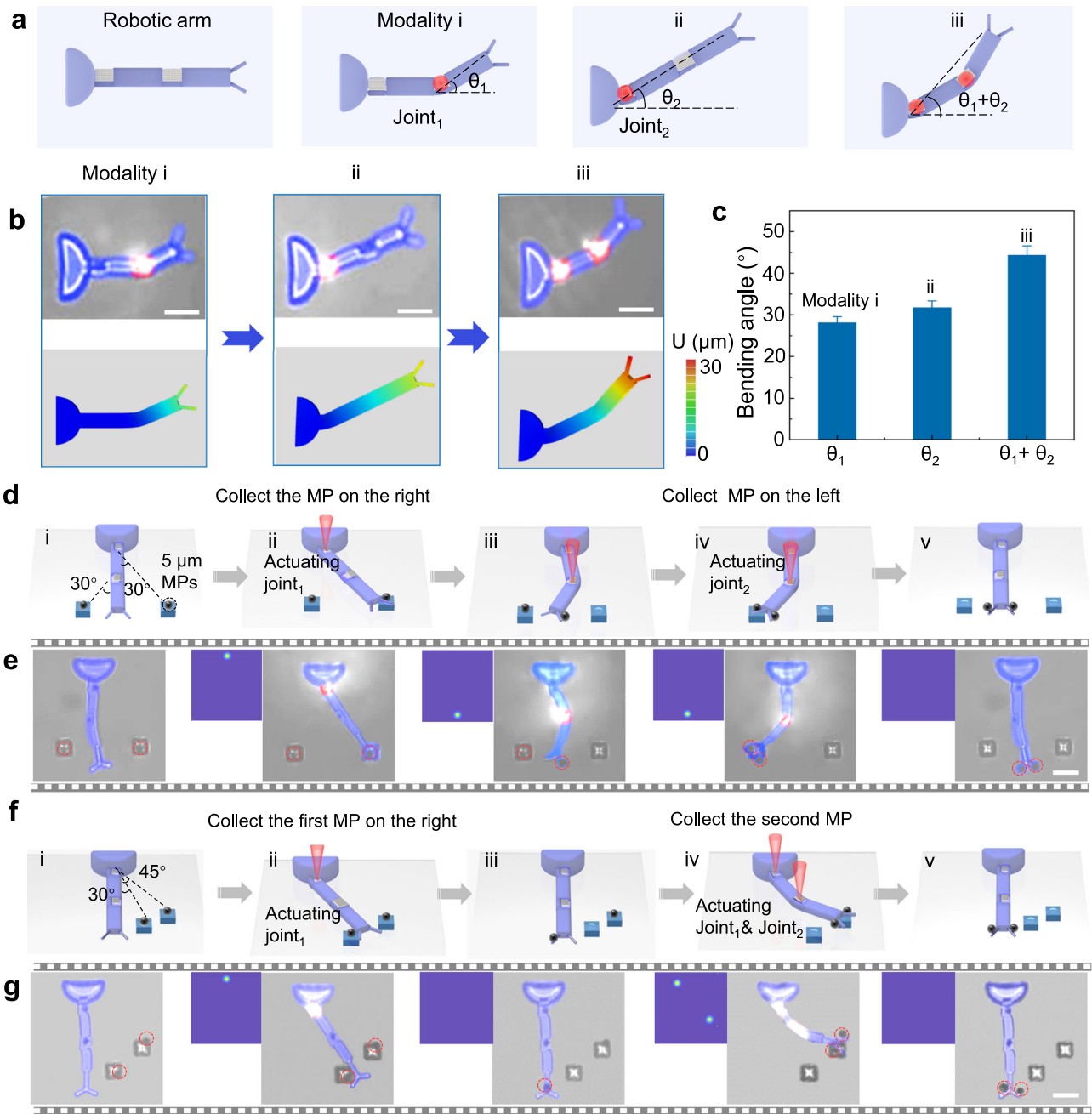

**Fig. 5 | A light-triggered multi-joint micro robotic arm (MRA). a** The schematic diagram of the morphing of the double-joint MRA stimulated by light-triggering. $\theta_1$ and $\theta_2$ represent the bending angle of joint$_1$ and joint$_2$, respectively. **b** The double-joint MRA is heated by different light modes to realize three modalities. **c** The total deformation of the end of the MRA gradually increases from 28° (modality i) to 45° μm (modality iii). **d** Schematic diagram and optical microscope images (**e**) of the MRA collecting two microparticles (MPs) at different locations on the opposite side. **f** Schematic diagram and optical microscope images (**g**) of the MRA collecting two MPs at different locations on one side. The collection works are both divided into five steps. All scale bars: 20 μm. Error bars stand for the standard error (*n* = 3).

K30 into 450 μL of ethylene glycol. 2 mg of Rhodamine 6 G was added for fluorescence imaging.

## Fabrication of 3D microstructures

A mode-locked Ti: sapphire ultrafast oscillator (Chameleon Vision-S, Coherent) with a ×60, NA = 1.35 oil immersion objective was used for fabrication. To increase the adhesion of the microstructure to the glass substrate, the coverslips were treated with a mixture that contained 3-Methacryloxypropyltrimethoxysilane and Methanol with a volume ratio of 1:19 for 12 h. The hydrogel precursor was first sonicated for 5 minutes for homogeneous mixing. Subsequently, 5 μL of the precursor solution was added dropwise to the coverslip, which was placed on a high-precision platform. The scanning oscillator in the XY plane and the Z-directional stage were controlled to achieve point-by-point scanning processing within the precursor solution. Point-by-point scanning paths were generated from STL models (SolidWorks) and converted to 3D point coordinate data by home-built software to command the combined X-Y-Z motion of the scanning mirror and piezoelectric stage. Subsequently, the processed sample was immersed in a developing solution (ethanol) for 15 min to remove the uncured hydrogel. Finally, we could observe the 3D-printed micro-actuators under the microscope after development. In the second step, the silver precursor was dropped on the structures and the selective deposition of Ag layers was reduced on the surface of

microstructures by the photoreduction process. Finally, the structures were washed with deionized water for light actuation. All of the light-triggered microactuators were actuated in deionized water.

## Dynamic multi-focal structured light fields generation

A reflection type liquid crystal SLM (Pluto NIR-II, from Holoeye Photonics AG, Germany) loaded with predesigned sequencing holograms was used to dynamically modulate the number and location of 800 nm laser beams (Chameleon Vision-S, Coherent). The holograms were calculated by the GS algorithm using MATLAB 2021a. The modulated beams were then projected through ×20 microscope objective (Olympus) onto the microstructures. A charge-couple device (CCD, MV-SUA134GM-T, Mindvison, China) camera was used to observe the workspace. A high-speed camera (2000 fps, Chronos 2.1-HD, Kron Technologies, Canada) was used to capture time-lapse images of the microplate to character the response time.

## Preparation of magnetic $SiO_2$ nanoparticles

The nanoparticle suspension contained the magnetic $SiO_2$ (purchased from Aladdin, M120194) with a diameter of 5 μm and deionized water with a volume ratio of 1:10, followed by 10 min ultrasonic mixing.

## Characterization of samples

The SEM images were taken with a secondary-electron scanning electron microscope (EVO18, ZEISS), and the samples were subjected to supercritical drying and coated with a gold film for 120 s beforehand. The optical microscopy images were obtained using an inverted microscope (DMI 3000B, Leica). The thickness and roughness of the Ag NPs layer were measured by a commercial Atomic Force Microscope (MFP-3D Origin, Asylum Research). The mechanical property (Young's modulus ($E_0$)) of hydrogels with different components were measured by a Micromechanical Testing System (FemtoTools, FT-MTA02). The rheological properties of the hydrogels were measured by a rotational rheometer (Physical MCR 301, Anton Paar, Austria).

## Simulation

The numerical simulation was performed in Comsol Multiphysics 5.4. The deformed process of the hybrid microstructures in response to the dynamic multifocal structured light fields involved large deformation and mass diffusion and heat transfer, which was proposed by He et al.[63]. Details of the simulation methods were reported in Supplementary Information. Multi-foci simulation was realized by Debye diffraction theory using MATLAB R2021a.

## Data availability

All data are available in the main text or the Supplementary Information. All data is also available from the corresponding author upon request.

## Code availability

All the relevant code in this paper is available upon request from the corresponding author.

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

## Acknowledgements

The authors acknowledge Dr. Jincheng Ni for his help in optical simulation. The authors acknowledge the Experimental Center of Engineering and Material Sciences at USTC for the fabrication and measuring of samples. This work was partly carried out at the USTC Center for Micro and Nanoscale Research and Fabrication. We thank Dr. Wenhao Zhang and Dr. Xiqi Wu for their help with AFM tests. We thank Prof. Wei Xiong and Dr. Zemin Zhang of Huazhong University of Science and Technology for their help with the mechanical tests. Parts of Fig. 1a were created with Biorender.com. The National Natural Science Foundation of China (Nos. 61927814, 52122511, 51875544, and 91963127), Major Scientific and Technological Projects in Anhui Province (201903a05020005), The Fundamental Research Funds for the Central Universities (WK2090000024), The China Postdoctoral Science Foundation (2021M703120), Open Research Fund of Advanced Laser Technology Laboratory of Anhui Province (No. AHL2020KF01).

## Author contributions

C.X. conceived the idea and designed the project. C.X., and Z.-G.R. performed all the experiments and the characterization. Z.-G.R and L.-R.Z. completed the simulation. C.X., Z.-G.R., D.-W.W., and D.W. completed data analysis and figure depicture. L.-R.Z., L.Y., Y.-L.H., J.-W.L., J.-R.C., and L.Z. discussed the results and made recommendations. All authors wrote and revised the paper. D.W. supervised the project.

## Competing interests

The authors declare no competing interests.
