## [Peer Review File · Nature Communications]

Light-triggered multi-joint microactuator fabricated by two-in-one femtosecond laser writingREVIEWER COMMENTS

Reviewer #1 (Remarks to the Author):

The paper, "Light-triggered multi-joint microactuator" reports a hydrogel based light responsive microstructure fabricated via 2-p laser fabrication. The bending microactuator is achieved by asymmetric polymerization (bilayer/gradient of crosslinking) and the light responsiveness is enabled by Ag nanoparticle clustering, induced by laser reduction within the same solution bath. A humanoid structure with four degree of freedom of deformation, demonstration of 3D movement, a snail actuator capable of collecting micro particles, are shown. The study also includes microscopy studies, finite element simulation, etc.

The novelties match the level of Nature Communication publication, in my opinion. Specifically, they are

1. The authors show a microscopic soft actuator system, less than a hundred micron, with programmable multi degree of freedom. The topic hits the important challenges in the field of soft robotics, regarding the miniaturization, remote control and degree of freedom of actuation.
2. By changing the laser writing parameter, the authors are able to control the crosslinking density during the 2-p polymerization, hence control the bending performance. Adding on top of it, they use laser induced reduction to include light absorbers at the selected location (on the joint). I believe each of these has been reported previously at macroscopic scale probably, but the combination of these at a truly microscopic scale, brings many new opportunities for robotics.
3. The biggest power of 2-p laser writing is to create 3D microscopic structure and movement. The authors show one preliminary demonstration in Fig. 4. I understand the difficulties in making such 3D actuation at a micro length scale, I won't ask for more about it.

Hopefully, the following comments can help improve the manuscript quality.

Main points.

1. I found it very misleading when the authors talk about joints, muscles, soft robotics, etc. "Joint" is a structure in conventional mechanical robot, from which people can define the term, "degree of freedom" (noted as deformation modalities, if I did not understand it in a wrong way). However, in the soft material based robot, i.e. the hydrogel actuator, the deformation is continuous, or say, the soft material contains numerous degree of freedom.

I understand the fact that the authors actually are building the joint structure by using a soft actuator, however, in the human arm analogue (Fig. 1), the authors refer hydrogel to human muscle (different from skeleton or joint), and then in the robotic design, the hydrogel is compared to both human muscle and joint.

I can easily understand the robotic design and concept, but, read from bioinspiration angle, I found those misleading.

2. 100 ms response time (or less as claimed), this important data is not supported by the data figure Fig. 2e. The data actually shows a value slightly larger than 100 ms. Or did I miss something?

Side notes.

1. Line 103, " where the contours of Ag NPs are marked in yellow." I guess the yellow marks are missing.

2. I disagree with "much lower than the timescale (1~10 s) required for other temperature-triggering³¹" in line 165. (i) Photothermal effect is still a temperature triggering process. (ii) The speeds recorded in different system, just because they are at different length scale. imagining that if the authors fabricated a millimeter sized hydrogel, the light induced actuation will then be slow.

3. What is "processing power" in line 71.

4. "However, due to the inherent limitations of structural design and fabrication methods, current light-triggered microactuators are still limited by a single deformation dimension (DD=1) and few deformation modalities (MT<3)." In line 57.

If we take the patterned light field driven soft actuator into account, the MT can be $\gg 3$ (Fischer, Nat. Mater. doi.org/10.1038/nmat4569). If we consider joint-contained structure (John Rogers, Sci. Rob. DOI: 10.1126/scirobotics.abn0602), in their case, DD=3, MT $\gg 3$.

5. Line 222, "Section 2.3"? An error due to reformatting, I assume.

6. Sadly, I have tried many software, but none of them can play the movies.

7. If the readers are not laser writing experts, they have difficulties in understand scanning time ratio, step length, scanning time. I wonder if the authors can give a easy explanation, since they are so important to determine the shrinkage/crosslinking density.

8. References: line 55, "exhibit precise local stimulation and short response times⁴³⁻⁴⁵", [45] is a paper about LCE not gel. For fast response, I recommend, <https://pubs.acs.org/doi/10.1021/acs.nanolett.7b00015> (this is not the reference from the reviewer) .

"For example, a pneumatic robot with multi-joint is expected to serve as a new generation of soft autonomous robots¹³." In [13] the autonomy is due to a gating system within the pneumatic system causing self-oscillation of gas generation, does not matter it is single joint or multi joint.

9. I assume the 10 mW is the input energy to the SLM, the focus spot size is unclear to me, thus I cannot estimate the intensity on the sample plane. Fig. S3 only shows the simulation results.

Reviewer #2 (Remarks to the Author):

In this work, the authors propose a multi-material 3D printing technique to fabricate micro joints and demonstrate a humanoid microactuator with multi-deformation modalities (>10), which only requires a short response time (<100 ms) and a low focal power (<10 mW) to realize a complete deformation. By designing the asymmetric porosity of hydrogel joints, various horizontal and vertical bending deformations are achieved in the resulting soft actuators. With the cooperation of multiple joints, the humanoid microactuator can perform several actions, such as raising hands, legs, and feet. Considering the highly contemporary significance of 4D printed soft microactuator and their promising applications, I would recommend this manuscript to be accepted for publication in Nature Communications with the following satisfactory improvement.

1. The authors should check if the silver nanoparticles are evenly distributed on the surface of the hydrogel, and whether they will be dislodged during multiple actuations.
2. The authors had better provide the thickness of the silver nanoparticle layer, and address the effects of the thickness on the photothermal effect.
3. Please provide the relevant mechanical property of PVP modified hydrogel, and whether PVP will affect the thermosensitivity of the hydrogel.
4. The authors had better provide the detailed 3D printing procedures of the thermo-responsive hydrogel on the coverslips, and the critical parameters of printing materials such as rheology and viscosity had better be provided.
5. The authors had better compare the properties of as-prepared soft actuators with those reported in the literatures.
6. The references below clearly related to bioinspired light-driven soft actuators and emerging applications had better be added, which would undoubtedly draw much more attentions of scientists and engineers from different backgrounds

Progress in Materials Science, 2021, 115, 100702; Advanced Materials, 2021, 33, 202004754; Materials Horizons, 2021, 8, 728; Advanced Functional Materials, 2022, 32, 2201884; Nano Today, 2022, 43, 101419; Materials Horizons, 2022, 9, 1825-1849; Angewandte Chemie International Edition, 2022, 61, e202211030.

Reviewer #3 (Remarks to the Author):

The manuscript demonstrates a humanoid hydrogel actuator with multiple joints to achieve multi-deformation modes. Despite of the clear illustration and systematic data collection and analysis, the manuscript is not distinguished enough to be published on

Nature Communication. Although the authors emphasize the short response time and low focal power required to achieve the deformation, it is not particularly noteworthy, given that the size of actuator is at micro scale. There is not much novelty in materials, fabrication process or structural design, thus not

commensurate with the level of other papers published on Nature Comm. Below are some specific comments:

1. It is not clear if the AgNPs are deposited in the hydrogel or on the surface of the hydrogel.
2. It is not clear whether the motion of the microactuator is triggered in water or in air.
3. Line 213, the subtitle should be “deformation in 3D space”.

Reviewer #1 (Remarks to the Author):

The paper, "Light-triggered multi-joint microactuator" reports a hydrogel based light responsive microstructure fabricated via 2-p laser fabrication. The bending microactuator is achieved by asymmetric polymerization (bilayer/gradient of crosslinking) and the light responsiveness is enabled by Ag nanoparticle clustering, induced by laser reduction within the same solution bath. A humanoid structure with four degree of freedom of deformation, demonstration of 3D movement, a snail actuator capable of collecting micro particles, are shown. The study also includes microscopy studies, finite element simulation, etc.

The novelties match the level of Nature Communication publication, in my opinion. Specifically, they are

1. The authors show a microscopic soft actuator system, less than a hundred micron, with programmable multi degree of freedom. The topic hits the important challenges in the field of soft robotics, regarding the miniaturization, remote control and degree of freedom of actuation.
2. By changing the laser writing parameter, the authors are able to control the crosslinking density during the 2-p polymerization, hence control the bending performance. Adding on top of it, they use laser induced reduction to include light absorbers at the selected location (on the joint). I believe each of these has been reported previously at macroscopic scale probably, but the combination of these at a truly microscopic scale, brings many new opportunities for robotics.
3. The biggest power of 2-p laser writing is to create 3D microscopic structure and movement. The authors show one preliminary demonstration in Fig. 4. I understand the difficulties in making such 3D actuation at a micro length scale, I won't ask for more about it.

Hopefully, the following comments can help improve the manuscript quality.

Response: Many thanks to the reviewer for careful review and high evaluation of this work. Your constructive comments have helped us to further improve our manuscript.

Main points.

1. I found it very misleading when the authors talk about joints, muscles, soft robotics, etc. "Joint" is

a structure in conventional mechanical robot, from which people can define the term, “degree of freedom” (noted as deformation modalities, if I did not understand it in a wrong way). However, in the soft material based robot, i.e. the hydrogel actuator, the deformation is continuous, or say, the soft material contains numerous degree of freedom.

I understand the fact that the authors actually are building the joint structure by using a soft actuator, however, in the human arm analogue (Fig. 1), the authors refer hydrogel to human muscle (different from skeleton or joint), and then in the robotic design, the hydrogel is compared to both human muscle and joint.

I can easily understand the robotic design and concept, but, read from bioinspiration angle, I found those misleading.

Response: Thanks for the reviewer’s valuable comments. We sincerely apologize for the confusions caused by the inappropriate descriptions. As the reviewer stated, the joint is generally a mechanical structure that could represent one degree of rotational freedom. In this work, we constructed micro-joint structures on soft microactuators so that the microactuator has multiple degrees of freedom of deformation. Upon careful consideration of the reviewer’s concerns, we agree that the comparison between hydrogel and human muscle in Fig. 1 is not appropriate. Therefore, we have revised the relevant schematic and description. Firstly, we only keep the joint part in the arm and remove the muscles in Fig. 1a to avoid misleading (Fig. R1.1). Correspondingly, we removed all descriptions of muscles, and the hydrogel micro-joints are only compared to the arm joints in the revised manuscript. We appreciate the reviewer’s insightful comments, which have helped us improve the clarity and accuracy of our work.

Fig. R1.1 (a) Bending deformation of human arm joints. (b) The schematic diagram of a light-triggered humanoid MJMA showing multiple modalities.

List of Revisions:

□ We revised the corresponding description on page 5 line 83: *“The flexible joint deformation of humans enables us to complete various modalities, such as walking, running, and jumping. Inspired by these joint bending behaviors (Fig. 1a), a photothermal-responsive hydrogel is chosen to construct flexible joints.”*

□ We deleted the “*muscle*” in page 5 line 89.

□ We revised Fig. 1a and its corresponding description on page 24 line 549: *“(a) The human arm relies on joint bending to achieve hand raising. (b)The schematic diagram of a light-triggered humanoid MJMA showing multiple modalities.”*

2. 100 ms response time (or less as claimed), this important data is not supported by the data figure Fig. 2e. The data shows a value slightly larger than 100 ms. Or did I miss something?

Response: We sincerely appreciate your comments. We apologize that the response time looks ~100 ms in Fig. 2e caused by a camera with low capture frames. As stated by the reviewer, short response time is important data. In the previous manuscript, the video was captured by a low frame rate camera (<20 fps), making it difficult to obtain an accurate value of the short response time. Therefore, we added the results of the hydrogel light-triggered deformation captured by a high-speed camera (2000 fps) to accurately evaluate the response time (Fig. R1.2). Different laser actuation powers (4~10 mW) are used to stimulate the microplate structure, respectively. And the frame-by-frame analysis shows that when the actuation power is 4 mW, the microstructure can reach a shrinkage rate of 0.19 after 60 ms. Meanwhile, when the actuation power increases to 10 mW, the shrinkage of the microstructure after 30 ms is ~0.35. Therefore, based on the above experiments and analysis, the light-triggering response time of the microplate is reduced to 30 ms with an actuation power of 10 mW. We correct the relevant Fig. 2e and add the microplate deformation results collected under the high-speed camera to the supplementary material Supplementary Fig. 15.

Fig. R1.2 Short response time characterization of hydrogel microplates. (a) Time-lapse images (0~60 ms) of microplates stimulated by different light actuation power (4 mW and 10 mW), captured by a high-speed camera (2000 fps). (b) Under the light actuation power of 4 mW and 10 mW, the deformation of the microplates is completed in 60 ms and 30 ms, respectively.

List of Revisions:

□ We added new photo-thermal response deformation images captured by high-speed camera (2000 fps) and revised the corresponding description on page 9 line 173: “When the AP increases from 4 mW to 10 mW, the response time of microplates deformation decreases from 60 ms to 30 ms (Fig. 2e, Supplementary Fig. 15, and Supplementary Movie 2).”

Page 16 line 350: “A high-speed camera (2000 fps, Chronos 2.1-HD, Kron Technologies, Canada) was used to capture time-lapse images of the microplate to character the response time.” was added.

□ We revised Fig. 2 and its corresponding description on page 25 line 567: “(e) As the local actuation laser power increases to 10 mW, the hydrogel demonstrates a short response time of 30 ms.”

□ We added Supplementary Fig. 15 in the supplementary materials and its corresponding description in the supplementary materials.

Side notes.

1. Line 103, ” where the contours of Ag NPs are marked in yellow.” I guess the yellow marks are missing.

Response: We thank the reviewer’s careful review. The yellow marks are missing in the enlarged SEM image. In the revised manuscript, we use the pseudo-color technique to mark the Ag NPs in SEM with

yellow color (Fig. R1.3).

Fig. R1.3 Light reduction of Ag NPs for photothermal conversion, in which silver ammonium ions absorb photons and are reduced to Ag NPs. SEM images show the corresponding states of hydrogel MJMAs (blue) and Ag NPs (yellow), respectively. Scale bar, 20 μm .

List of Revisions:

□ We revised Fig. 1d and its corresponding description on page 24 line 554: “*SEM images show the corresponding materials of hydrogel MJMAs (blue) and Ag NPs (yellow), respectively.*”

2. I disagree with “much lower than the timescale (1~10 s) required for other temperature-triggering³¹” in line 165. (i) Photothermal effect is still a temperature triggering process. (ii) The speeds recorded in different system, just because they are at different length scale. imagining that if the authors fabricated a millimeter sized hydrogel, the light induced actuation will then be slow.

Response: Thank you for the reviewer’s comment. We agree with the reviewers’ statements that the photothermal effect is still a temperature-triggering process. And the response time is also related to the scale of the system size, that is, the smaller the size, the faster the response time. Here, we would like to express that temperature-triggering relies on temperature switching of the global liquid environment. For example, a flower-like microgel actuation in response to temperature (20 to 32 $^{\circ}\text{C}$ and then 32 to 20 $^{\circ}\text{C}$) takes several minutes due to the slow liquid temperature switch¹. In contrast, light-triggering is a mode that enables fast temperature switching in localized regions, leading to a reduction of the temperature transition time to 30 ms in localized regions surrounding the microactuator. Therefore, compared to the temperature-triggering mode that relies on global liquid heating and cooling, the local light-triggering could rapidly switch the temperature of local regions around microstructures, resulting in the microstructure exhibiting shorter deformation time. To prevent misunderstanding among the readers, we have modified the original statement.

Reference

1. Nishiguchi, A., Mourran, A., Zhang, H. & Moller, M. In-gel direct laser writing for 3d-designed hydrogel composites that undergo complex self-shaping. *Adv. Sci.* 5, 1700038 (2018).

List of Revisions:

□ We revised the corresponding description on page 9 line 175: *“Compared to the temperature-triggering mode that depends on the heating and cooling of the surrounding liquid⁶⁷, the local light-triggering could rapidly switch the temperature of local regions surrounding the microstructures, resulting in the microstructure exhibiting shorter deformation time.”*

3. What is “processing power” in line 71.

Response: We apologize for bringing the misunderstanding to the reviewer. In our work, the fabrication and actuation of microactuators are realized in the same femtosecond (Fs) laser system. Therefore, we define the laser power for fabricating microactuators as the processing power and the laser power for microactuator deformation as the actuation power. We revised “processing power” to “processing laser power” and added its explanation in the corresponding sections of the manuscript to help readers better understand.

List of Revisions:

□ We revised the corresponding description on page 4 line 73: *“even lower than the processing laser power (33 mW) of the microactuator.”*

□ We defined the processing laser power on page 8 line 152: *“DLW is a point-by-point scanning process so that the 3D microstructure is built up from one focal spot to another^{65, 66}. In this way, PLP is the power of the focal spot, SPS is the distance between two adjacent focal spots, and ST is the dwell time of the focal spot at each position.”*

4. “However, due to the inherent limitations of structural design and fabrication methods, current light-triggered microactuators are still limited by a single deformation dimension (DD=1) and few deformation modalities (MT<3).” In line 57.

If we take the patterned light field driven soft actuator into account, the MT can be $\gg 3$ (Fischer, Nat. Mater. doi.org/10.1038/nmat4569). If we consider joint-contained structure (John Rogers, *Sci. Rob.*

DOI: 10.1126/scirobotics.abn0602), in their case, DD=3, MT>>3.

Response: We thank and agree with the reviewer's statement. According to the references proposed by the reviewer, the light-triggered soft actuator based on liquid crystal elastomer indeed could achieve swimming and versatile locomotion by 2D structured light (Fischer, [Nat. Mater. doi.org/10.1038/nmat4569](https://doi.org/10.1038/nmat4569)). Besides, terrestrial robots with multiple shape memory alloy joints also could complete multiple deformation modes by scanning light (John Rogers, Sci. Rob. DOI: 10.1126/scirobotics.abn0602). However, precisely tuning the light distribution in 3D space to independently stimulate each local joint deformation of the microactuator remains challenging. In addition, the microactuators mentioned above have the potential to continue to be miniaturized. In our study, the local deformation of the MJMAs can be precisely tuned in 3D space by multi-foci structured light. And we have further reduced the size of the microactuators (<100 μm). Based on the reviewer's comment, we have corrected the previous inappropriate description.

List of Revisions:

□ We revised the corresponding description on page 4 line 56: *“Recently, light-triggering as a flexible and remote-control mode is widely used to drive soft actuators^{50, 51}. Advanced 2D structured light and dynamic light scanning methods promote the miniaturized actuators to increase their deformation modalities^{18, 52} (Supplementary Table 1). However, precisely tuning the light distribution to independently stimulate the local region deformation of the 3D microactuator (<100 μm) for achieving multi-modality in 3D space remains challenging. Building multiple joints in microactuators and achieving independent control of each joint is an attractive way to increase the microactuator's modalities and manipulation capabilities.”*

5. Line 222, “Section 2.3”? An error due to reformatting, I assume.

Response: We apologize for the errors caused by the formatting. Section 2.3 presents the humanoid microactuator that deforms in a single XY plane. We revised the related description to correct the mistake.

List of Revisions:

□ We revised the corresponding description on page 12 line 240: *“it is worth noting that the multi-foci*

here are generated in 3D space."

6. Sadly, I have tried many software, but none of them can play the movies.

Response: We are sorry for hearing about that the reviewer cannot watch the movies. In the revised version, we convert all movies to MP4 format for reviewer to view.

7. If the readers are not laser writing experts, they have difficulties in understand scanning time ratio, step length, scanning time. I wonder if the authors can give a easy explanation, since they are so important to determine the shrinkage/crosslinking density.

Response: Thanks for the reviewer's kind advice. As the reviewer stated, it is difficult for non-laser-oriented experts to understand some of the relevant terminologies. Therefore, we have added explanations of the relevant terms and schematic to help the general audience understand. For direct laser writing, it is a point-by-point scanning process, so the 3D microstructure is built up from one scanning point to another. As shown in Fig. R1.4, the laser processes a series of point voxels along the scanning path, so the processing laser power (PLP) is the power of the laser focal spot, the scanning time (ST) is the dwell time of the laser focal spot at each scanning point position. In addition, the distance between two adjacent scanning point voxels is defined as the scanning point step (SPS). The exposure dose (ED) per unit volume is the main factor affecting the degree of crosslinking and shrinkage of the hydrogel, which can be considered as,

$$ED = n * P * T \tag{1}$$

Where n, P, and t are the number of scanning points per unit volume, the PLP per point, and the ST per point. Therefore, when other conditions are constant, the longer the ST or the larger the PLP at a single point, the higher the degree of cross-linking and the smaller the shrinkage of the hydrogel. In addition, the larger the SPS, the smaller the number of scanning points, which eventually leads to a decrease in the degree of crosslinking and an increase in shrinkage.

Fig. R1.4 The process and parameters in direct laser writing (DLW) for microactuator fabrication. **List of Revisions:**

□ We revised the corresponding description on page 8 line 149: “Subsequently, to investigate the shrinkage property of hydrogel, we explore the effect of direct laser writing (DLW) parameters on hydrogel shrinkage including processing laser power (PLP), scanning point step (SPS), and scanning time (ST). DLW is a point-by-point scanning process so that the 3D microstructure is built up from one focal spot to another^{65, 66}. In this way, PLP is the power of the focal spot, SPS is the distance between two adjacent focal spots, and ST is the dwell time of the focal spot at each position.”

□ We revised Supplementary Fig. 1 in the supplementary materials and added the corresponding description on page 16 line 330: “The hydrogel precursor was first sonicated for 5 minutes for homogeneous mixing. Subsequently, 5 μL of the precursor solution was added dropwise to the coverslip, which was placed on a high-precision platform. The scanning oscillator in the XY plane and the Z-directional stage were controlled to achieve point-by-point scanning processing within the precursor solution. Point-by-point scanning paths were generated from STL models (SolidWorks) and converted to 3D point coordinate data by home-built software to command the combined X-Y-Z motion of the scanning mirror and piezoelectric stage. Subsequently, the processed sample was immersed in a developing solution (ethanol) for 15 minutes to remove the uncured hydrogel. Finally, we could observe the 3D-printed microactuators under the microscope after development.”

8. References: line 55, “exhibit precise local stimulation and short response times^{43 -45}”, [45] is a paper about LCE not gel. For fast response, I recommend,

<https://pubs.acs.org/doi/10.1021/acs.nanolett.7b00015> (this is not the reference from the reviewer).

“For example, a pneumatic robot with multi-joint is expected to serve as a new generation of soft autonomous robots¹³.” In [13] the autonomy is due to a gating system within the pneumatic system causing self-oscillation of gas generation, does not matter it is single joint or multi joint.

Response: Thanks for the reviewer’s careful review and nice advice. To match the information described in line 55, we added the literature² recommended by the reviewers to the reference list. Besides, we agree with the reviewer’s statement about the pneumatic robot we listed. To better describe the advantages of multimodal soft robots, we have added a new example. For example, terrestrial robots with multiple shape memory alloy joints can realize linear/curvilinear crawling, walking, turning, and jumping by laser-inducing³. Based on the reviewer’s comments, we have revised the relevant descriptions in the manuscript.

References

2. Zhang, H., Mourran, A. & Moller, M. Dynamic switching of helical microgel ribbons. *Nano Lett.* 17, 2010-2014 (2017).

3. Han, M. et al. Submillimeter-scale multimaterial terrestrial robots. *Sci. Robot.* 7, eabn0602 (2022). **List of Revisions:**

□ We revised the corresponding description on page 3 line 36: “*For example, terrestrial robots with multiple shape memory alloy joints can realize linear/curvilinear crawling, walking, turning, and jumping by laser-inducing¹⁸.*”

□ We added new references on page 19 line 409 and page 21 line 470.

9. I assume the 10 mW is the input energy to the SLM, the focus spot size is unclear to me, thus I cannot estimate the intensity on the sample plane. Fig. S3 only shows the simulation results.

Response: Thank you for the reviewer’s good questions. To facilitate the reviewer to evaluate the laser intensity on the microactuator plane, we have added the corresponding theoretical simulation and experiment. In our work, the actuation laser power (10 mW) is the power allocated to each focal spot in multi-foci mode. According to the Debye diffraction simulation, we found that the diameter of the multifocal point after focusing under the objective lens is 2.269 μm . Correspondingly, the measured results visually demonstrate the diameter of the focus spot is $\sim 2.3 \mu\text{m}$ (Figure R1.5). It can be seen that

the measured results are in good agreement with the simulation results, and the multifocal diameter and field strength are uniformly distributed on the focal plane.

Finally, the intensity I can be estimated with the following formula:

$$= \left(\frac{2}{\lambda} \right)^2 \tag{R(3)}$$

Therefore, the average density of focus spot is calculated as $2.41 \text{ mW} \cdot \mu\text{m}^{-2}$.

Fig. R1.5 Density simulation and measurement of four foci modulated by the GS algorithm, where the diameter of each light foci is about $2.3 \mu\text{m}$.

List of Revisions:

- We revised Supplementary Fig. 5 in the supplementary materials and the corresponding description on page 6 line 111: “*The measured multi-foci spatial position and intensity are consistent with the simulated results, where the diameter of each light foci is $\sim 2.3 \mu\text{m}$ (Supplementary Fig. 5).*”

Reviewer #2 (Remarks to the Author):

In this work, the authors propose a multi-material 3D printing technique to fabricate micro joints and demonstrate a humanoid microactuator with multi-deformation modalities (>10), which only requires a short response time (<100 ms) and a low focal power (<10 mW) to realize a complete deformation. By designing the asymmetric porosity of hydrogel joints, various horizontal and vertical bending deformations are achieved in the resulting soft actuators. With the cooperation of multiple joints, the humanoid microactuator can perform several actions, such as raising hands, legs, and feet. Considering the highly contemporary significance of 4D printed soft microactuator and their promising applications, I would recommend this manuscript to be accepted for publication in Nature Communications with the following satisfactory improvement.

Response: Thank you very much for your positive comments. We have answered each comment and made the appropriate changes in the revised version. Please review the details of the point-by-point responses below.

1. The authors should check if the silver nanoparticles are evenly distributed on the surface of the hydrogel, and whether they will be dislodged during multiple actuations.

Response: Many thanks to the reviewer's valuable comments. To check the state of Ag NPs on the hydrogel surface, we tested the Ag NPs layer with atomic force microscopy (AFM). As shown in Fig. R2.1, a layer of Ag NPs is deposited on the surface of the hydrogel. Three areas ($3\ \mu\text{m} \times 3\ \mu\text{m}$) are taken in the Ag NPs layer and their roughness (root mean square RMS) are measured as 52.4 nm, 67.6 nm, and 55.5 nm, respectively. In addition, the thickness information of the three Ag NPs extraction lines shows that the thicknesses of the three lines are concentrated between 300 nm and 400 nm. Thus, we consider that Ag NPs are uniformly distributed on the surface of the hydrogel.

In addition, we also check the distribution of Ag NPs on the hydrogel surface after multiple actuations. There is no displacement of Ag NPs after multiple actuation tests, and they are still uniformly distributed on the hydrogel surface (Fig. R2.1d~f). Therefore, the microactuator demonstrate stable shrinking ratio under 100 times actuation.

Fig. R2.1 Characterization of thickness and roughness of Ag NPs layer. (a) AFM scanned a 3D image of the Ag NPs layer. (b) The roughness of the three areas of the Ag NPs layer are 52.4 nm, 67.6 nm, and 55.5 nm, respectively. (c) The thickness of the Ag NPs layer on the three extraction lines is between 300 nm and 400 nm. (d) The microscopic images of the microplate in expanding and shrinking state in the 1st, 50th, and 100th cycle. (e) The SEM image of Ag NPs layer deposited on the surface of hydrogel after 100 times actuation. (f) The microplate exhibits stable shrinking ratio under 100 cycles.

List of Revisions:

- We added the corresponding description on page 6 line 95: *“In this step, the hydrogel structure is immersed in an Ag ink, and silver ammonium ions are reduced to Ag NPs by absorbing photons, which are uniformly attached to the hydrogel joints with average roughness of 58.5 nm (Supplementary Fig. 3).”*
- Page 9 line 180: *“In addition, according to the optical and SEM images, no Ag NPs are dislodged during multiple actuations (Supplementary Fig. 16 and Supplementary Movie 3).”* was added.
- We added Supplementary Fig. 3, Supplementary Fig. 16, Supplementary Movie 3, and the corresponding description in the supplementary materials.

2. The authors had better provide the thickness of the silver nanoparticle layer, and address the effects

of the thickness on the photothermal effect.

Response: Thank you for the reviewer's great advice. To characterize the thickness of the Ag NPs layer, we supplemented the relevant AFM experiments. In our study, the thickness of Ag NPs is positively correlated with the power of laser reduction. As shown in Fig. R2.2 (a~b), Ag NPs have different thicknesses at different laser reduction powers, from 189 nm at 2 mW to 464 nm at 5 mW. In addition, Different thicknesses of Ag NPs have a certain effect on the photothermal effect and further affect the deformation properties of the hydrogel microstructure. Fig. R2.2 (c~d) show that when the thickness of Ag NPs is insufficient (<200 nm), the shrinking rate of the hydrogel is small (~0.29). And when Ag NPs reach 300 nm and above, the shrinking ratio of the microplate is ~0.35, which reaches the maximum shrinking rate of the hydrogel.

Fig. R2.2 The effects of the Ag NPs thickness on the photothermal effect. (a) AFM images of Ag NPs layers fabricated with different laser reduction powers. (b) The layer thickness of Ag NPs increases from 189 nm to 464 nm with the increase of laser reduction power (2 mW~5 mW). (c) Photothermal shrinkage microscope images of microplates with different thicknesses of Ag NPs layer. (d) When the layer thickness of Ag NPs is less than 200 nm, the shrinkage of the microplate is only 0.293. When the layer thickness of Ag NPs is higher than 300 nm, the shrinkage rate of hydrogel reaches the maximum value (~0.35).

List of Revisions:

- We added the corresponding description on page 8 line 166: *“In addition to the processing parameters of the hydrogel, the thickness of the Ag NPs layer produced by laser reduction has an important effect on the shrinking ratio of light-triggered microstructure. When the layer thickness of Ag NPs is less than 200 nm, the shrinking ratio of the microplate is 0.29. When the layer thickness of Ag NPs is higher than 300 nm, the shrinking ratio of hydrogel reaches the maximum value (Supplementary Fig. 13).”*
- Page 17 line 359: *“The thickness and roughness of the Ag NPs layer were measured by a commercial Atomic Force Microscope (MFP-3D Origin, Asylum Research).”*
- We added Supplementary Fig. 13 and the corresponding description in the supplementary materials.

3. Please provide the relevant mechanical property of PVP modified hydrogel, and whether PVP will affect the thermosensitivity of the hydrogel.

Response: Thanks for the reviewer’s valuable comments. According to the reviewer’s advice, we added a series of experiments to refine the manuscript. Firstly, we test the mechanical property (Young’s modulus (E_0)) of hydrogels with different components by a Micromechanical Testing System (FemtoTools, FT-MTA02). An FT-S100000 microforce sensing probe with a force range of $\pm 100,000$ μN and a resolution of 5 μN is used with a $50 \mu\text{m} \times 50 \mu\text{m}$ tip to measure the compression forces of blocks of polymerized hydrogel with dimensions of $40 \mu\text{m} \times 40 \mu\text{m} \times 15 \mu\text{m}$ (Fig. R2.3a). Hydrogels (with and without PVP) are printed with the same laser scanning parameters (PLP=33 mW, ST=3 ms). Subsequently, the pressure vs displacement curves of the hydrogels are measured (Fig. R2.3b). The E_0 could be calculated as,

$$E_0 = P/e = \frac{F/S}{\Delta L/L} \quad \text{R (2.1)}$$

Where $S=1.6 \times 10^{-9} \text{ m}^2$, $L=1.5 \times 10^{-5} \text{ m}$, $\Delta L=10^{-6} \text{ m}$ are the area, height, and compression displacement of the microplate. F is the pressure loaded on the microplate (327 μN and 676 μN of hydrogels with and without PVP addition).

Thus, compared with the hydrogel microstructure without PVP addition (3.07 Mpa), the hydrogel microstructure with PVP addition (5% wt) has a higher E_0 of 6.34 Mpa. It was also evident that processing microstructures in hydrogels without PVP addition led to significant defects during

processing. On the contrary, the PVP-added hydrogel can print high-quality 3D microactuators (Fig. R2.3 (c)). In addition, the laser scanning parameters also significantly affect the E_0 of the hydrogel (with PVP). In our study, the higher the crosslink density of the hydrogels when the laser scan time increases, which leads to an increase in their mechanical properties. Therefore, we test three groups of hydrogels (with PVP) with different laser scanning times (1 ms, 2 ms, and 3 ms), which have F of 165 μN , 351 μN , and 676 μN , resulting in E_0 of 1.55 Mpa, 3.29 Mpa, and 6.34 Mpa, respectively (Fig. R2.3 (d)).

Fig. R2.3 Mechanical test of hydrogels with different components. (a) Setup of a Micromechanical Testing System. (b) Mechanical test of hydrogels with and without PVP additions. (c) Compared to hydrogel without PVP, hydrogel with PVP demonstrates stronger structural properties. (d) Mechanical test of hydrogels (with PVP) with different laser scanning times (1~3 ms).

In addition, we check the effect of PVP on the thermosensitivity of hydrogels. As shown in Fig. R2.4, compared to the hydrogel without PVP addition with the shrink ratio of ~ 0.37 , the microstructure shrinking rate of the hydrogel with PVP addition is slightly reduced to ~ 0.35 . The thermal response curve of the hydrogel with PVP addition remains the same as that of the hydrogel without PVP. More

importantly, the light-triggered shrinking ability of hydrogel with PVP addition is comparable to that of hydrogels without PVP addition. Thus, PVP significantly enhances the mechanical properties of hydrogels while having essentially no negative impact on the thermosensitivity and photothermal sensitivity of hydrogels.

Fig. R2.4 Effect of PVP on the thermal sensitivity of hydrogels. **(a)** Time-lapse images of shrinkage of hydrogel microplates with and without PVP addition with increasing temperature. **(b)** The thermal response characteristics of hydrogel microstructures before and after the addition of PVP. **(c)** Photothermal response properties of hydrogels with PVP addition, where shrinkage and response times are comparable to those of hydrogels without PVP addition. Scale bars, 10 μm.

List of Revisions:

□ We revised the corresponding description on page 7 line 131: “*The PVP significantly enhances the mechanical properties of hydrogel microstructures (Supplementary Fig. 7) while having essentially no negative impact on the thermosensitivity and photothermal sensitivity of the hydrogel (Supplementary Fig. 8). In addition, the addition of PVP significantly increases the viscosity of the hydrogel precursor*

and prevents low processing quality due to the flow of the hydrogel during processing (Supplementary Fig. 9).”

□ We added description of mechanical test on page 17 line 360: *“The mechanical property (Young’s modulus (E_0)) of hydrogels with different components were measured by a Micromechanical Testing System (FemtoTools, FT-MTA02).”*

□ We added Fig Supplementary Fig. 7~9 and the corresponding description in the supplementary materials.

4. The authors had better provide the detailed 3D printing procedures of the thermo-responsive hydrogel on the coverslips, and the critical parameters of printing materials such as rheology and viscosity had better be provided.

Response: Thank you for your comments. To show the reader the printing process more clearly, we have added a diagram and a more detailed description (Fig. R2.5). The hydrogel precursor was first sonicated for 5 min for homogeneous mixing. Subsequently, 5 μL of the precursor solution was added dropwise to the coverslip, which was placed on a high-precision platform. The scanning oscillator in the XY plane and the Z-directional stage are controlled to achieve point-by-point scanning processing within the precursor solution. Point-by-point scanning paths are generated from STL models (SolidWorks) and converted to 3D point coordinate data by home-built software to command the combined x-y-z motion of the scanning mirror and piezoelectric stage. 3D hydrogel microstructures are built by the accumulation of scanning points as the laser focal spot follows the scan path from one scanning point to another. Subsequently, the processed sample was immersed in a developing solution (ethanol) for 15 minutes to remove the unpolymerized hydrogel. Finally, we could observe the 3D-printed microactuators under the microscope after development.

Fig. R2.5 The detailed 3D printing procedures of the thermo-responsive hydrogel on the coverslips. **List of Revisions:**

□ We revised Supplementary Fig. 1 in the supplementary materials and added the corresponding description on page 16 line 330: *“The hydrogel precursor was first sonicated for 5 minutes for homogeneous mixing. Subsequently, 5 μ L of the precursor solution was added dropwise to the coverslip, which was placed on a high-precision platform. The scanning oscillator in the XY plane and the Z-directional stage were controlled to achieve point-by-point scanning processing within the precursor solution. Point-by-point scanning paths were generated from STL models (SolidWorks) and converted to 3D point coordinate data by home-built software to command the combined X-Y-Z motion of the scanning mirror and piezoelectric stage. Subsequently, the processed sample was immersed in a developing solution (ethanol) for 15 minutes to remove the uncured hydrogel. Finally, we could observe the 3D-printed microactuators under the microscope after development..”*

In addition to detailed 3D printing procedures, based on the reviewer’s suggestion, we added experiments to test the rheology and viscosity of the hydrogel material. As shown in Fig. R2.6, The shear force versus shear rate curve proves that the fluids are Newtonian fluids. And the stable viscosity of hydrogel increases from 0.024 Pa s without PVP addition to 0.073 Pa·s with PVP addition, respectively. In our work, the addition of PVP significantly increases the viscosity of the hydrogel precursor and prevents low processing quality due to the flow of the hydrogel during processing.

Figure
R2.6

Rheology and viscosity testing of prepared hydrogels (All experiments are tested under 25°C). The viscosity of hydrogel precursor with PVP addition is 0.073 Pa·s, higher than that of hydrogel without PVP addition (0.024 Pa·s).

List of Revisions:

- We revised the corresponding description on page 7 line 134: *“In addition, the addition of PVP significantly increases the viscosity of the hydrogel precursor (0.073 Pa·s) to prevent low processing quality due to the flow of the hydrogel during processing (Supplementary Fig. 9).”*
- Page 17 line 362: *“The rheological properties of the hydrogels were measured by a rotational rheometer (Physical MCR 301, Anton Paar, Austria).”* was added.
- We added Fig S9 and the corresponding description in the supplementary materials.

5. The authors had better compare the properties of as-prepared soft actuators with those reported in the literatures.

Response: Thank you for the reviewer’s great advice. To provide the state-of-the-art of multi-joint microactuators, modalities numbers, the size of actuators, used materials, response time, and stimuli mode of previously reported soft actuators have been comprehensively summarized in a table (Table R2.1) below. Compared to the existing soft actuators, our multi-joint microactuators fabricated by two-in-one Fs laser printing demonstrate multiple modality numbers (>10), much smaller 3D structure sizes (<100 μm), low actuation power (10 mW), and short response time (30 ms). Furthermore, thanks to the combination of a structured multi-foci beam and a multi-joint microactuator, it can operate multiple micro cargos in different locations, which is similar to a multi-joint robot arm in a factory. All in all,

we believe that light-triggered multi-joint actuators at a truly microscopic scale will bring many new opportunities for soft robotics, such as microcargo handling, precision assembly, and microdevice integration. To distinguish our microactuators from others, a table has been added to the supporting information (Table S1). And the relevant discussion has been added to the revised version.

Soft actuators	*Modality numbers	Size	Structure	Respose time	Response Materials	Stimuli mode
Ref. 4	2	>1 cm	3D	>1 min	Hydrogel	Water
Ref. 5	2	>2 mm	2D	>1.5 min	Hydrogel	pH
Ref. 6	2	~100 μm	3D	NA	Hydrogel	pH
Ref. 7	2	>1 mm	2D	NA	Hydrogel	Magnetic field
Ref. 8	2	>1 cm	2D	>1 min	Hydrogel	Temperature
Ref. 1	2	~100 μm	3D	>1 min	Hydrogel	Temperature
Ref. 9	2	>1 mm	2D	NA	Hydrogel	Temperature
Ref. 10	2	~100 μm	3D	~100 ms	Hydrogel	Temperature & light
Ref. 11	2	>1 cm	2D	>1 s	Hydrogel	Light (175~900 mW)
Ref. 2	2	< 100 μm	2D	<100 ms	Hydrogel	Light (2.5 W)
Ref. 12	2	<100 μm	3D	300 ms	Hydrogel	Light (70~120 mW)
Ref. 13	4	>2 cm	2D	>2 s	Liquid crystal elastomer	Different Lights (520, 655, 808 nm) 100~300 mW
Ref. 14	>10	>200 μm	2D	NA	Liquid crystal elastomer	2D structured light (2.5 W)
Ref. 3	>10	>500 μm	2D to 3D	NA	Shape memory alloy	2D scanning light (188 mW)
This work	>10	<100 μm	3D	30 ms	Hydrogel	3D Structured light (<10 mW)

Table R2.1 Comparison of current soft actuators.

*Modalities numbers: Here we only consider the number of modalities in the steady state.

References

1. Nishiguchi, A., Mourran, A., Zhang, H. & Moller, M. In-gel direct laser writing for 3D-designed hydrogel composites that undergo complex self-shaping. *Adv. Sci.* **5**, 1700038 (2018).
2. Zhang, H., Mourran, A. & Moller, M. Dynamic switching of helical microgel ribbons. *Nano Lett.* **17**, 2010-2014 (2017).

3. Han, M. et al. Submillimeter-scale multimaterial terrestrial robots. *Sci. Robot.* **7**, eabn0602 (2022).
4. Sydney Gladman, A., Matsumoto, E.A., Nuzzo, R.G., Mahadevan, L. & Lewis, J.A. Biomimetic 4d printing. *Nat. Mater.* **15**, 413-418 (2016).
5. Li, H., Go, G., Ko, S.Y., Park, J.O. & Park, S. Magnetic actuated pH-responsive hydrogel-based soft micro-robot for targeted drug delivery. *Smart Mater. Struct.* **25**, 027001 (2016).
6. Li, R. et al. Stimuli-responsive actuator fabricated by dynamic asymmetric femtosecond bessel beam for in situ particle and cell manipulation. *ACS Nano* **14**, 5233-5242 (2020).
7. Goudu, S.R. et al. Biodegradable untethered magnetic hydrogel milli-grippers. *Adv. Funct. Mater.* **30**, 2004975 (2020).
8. Pantula, A. et al. Untethered unidirectionally crawling gels driven by asymmetry in contact forces. *Sci. Robot.* **7**, eadd2903 (2022).
9. Malachowski, K. et al. Stimuli-responsive theragrippers for chemomechanical controlled release. *Angew. Chem. Int. Ed.* **53**, 8045-8049 (2014).
10. Hippler, M. et al. Controlling the shape of 3d microstructures by temperature and light. *Nat. Commun.* **10**, 232 (2019).
11. Zhao, Y. et al. Soft phototactic swimmer based on self-sustained hydrogel oscillator. *Sci. Robot.* **4**, eaax7112 (2019).
12. Deng, C.S. et al. Femtosecond laser 4d printing of light-driven intelligent micromachines. *Adv. Funct. Mater.* **33**, 2211473 (2023).
13. Yang, X. et al. Bioinspired light-fueled water-walking soft robots based on liquid crystal network actuators with polymerizable miniaturized gold nanorods. *Nano Today* **43**, 101419 (2022).
14. Palagi, S. et al. Structured light enables biomimetic swimming and versatile locomotion of photoresponsive soft microrobots. *Nat. Mater.* **15**, 647-653 (2016).

List of Revisions:

□ We added the corresponding description on page 11 line 227: “To provide the state-of-the-art of MJMAs, soft actuators have been comprehensively summarized in Supplementary Table 1. Compared to the existing soft actuators, our microactuators constructed by two-in-one Fs printing demonstrate

multiple modality numbers (>10), much smaller 3D structure sizes (<100 μm), precise local region response (30 ms), and low actuation power (<10 mW).”

□ We added the supplementary Table 1 and the corresponding references in the supplementary materials.

6. The references below clearly related to bioinspired light-driven soft actuators and emerging applications had better be added, which would undoubtedly draw much more attentions of scientists and engineers from different backgrounds

Progress in Materials Science, 2021, 115, 100702; Advanced Materials, 2021, 33, 202004754; Materials Horizons, 2021, 8, 728; Advanced Functional Materials, 2022, 32, 2201884; Nano Today, 2022, 43, 101419; Materials Horizons, 2022, 9, 1825-1849; Angewandte Chemie International Edition, 2022, 61, e202211030.

Response: We thank and accept the reviewer’s kind advice. Thanks to the reviewers for listing a range of works on liquid crystal elastomers and hydrogels, we have added these important works to the references to attract the attention of scientists from different backgrounds.

Added references

1. Yang, X. et al. Bioinspired light-fueled water-walking soft robots based on liquid crystal network actuators with polymerizable miniaturized gold nanorods. *Nano Today* **43**, 101419 (2022).
2. Tang, L. et al. Poly(n-isopropylacrylamide)-based smart hydrogels: Design, properties and applications. *Prog. Mater. Sci.* **115**, 100702 (2021).
3. Yang, J. et al. Beyond the visible: Bioinspired infrared adaptive materials. *Adv. Mater.* **33**, e2004754 (2021).
4. Lv, P. et al. Stimulus-driven liquid metal and liquid crystal network actuators for programmable soft robotics. *Mater. Horiz.* **8**, 2475-2484 (2021).
5. Yang, M.Y. et al. Bioinspired phototropic mxene-reinforced soft tubular actuators for omnidirectional light-tracking and adaptive photovoltaics. *Adv. Funct. Mater.* **32**, 2201884 (2022).
6. Guan, Z., Wang, L. & Bae, J. Advances in 4D printing of liquid crystalline elastomers: materials, techniques, and applications. *Mater. Horiz.* **9**, 1825 (2022).
7. Zhang, X. et al. Three-dimensional electrochromic soft photonic crystals based on mxene-

integrated blue phase liquid crystals for bioinspired visible and infrared camouflage. *Angew. Chem. Int. Ed.* **61**, e202211030 (2022).

Reviewer #3 (Remarks to the Author):

The manuscript demonstrates a humanoid hydrogel actuator with multiple joints to achieve multi-deformation modes. Despite of the clear illustration and systematic data collection and analysis, the manuscript is not distinguished enough to be published on Nature Communication. Although the authors emphasize the short response time and low focal power required to achieve the deformation, it is not particularly noteworthy, given that the size of actuator is at micro scale.

Response: We sincerely thank the reviewer for the valuable comments. Your thoughtful comments have helped me to refine the manuscript and articulate my arguments more clearly and convincingly. In response to the reviewer's concern regarding the significance of the short response time and low actuation power, we would like to highlight their importance for potential future applications. For example, short response time is critical when microactuators need to manipulate fast-moving micro-objects (*e.g.*, microparticles and cells) in fluids. Furthermore, low actuation power also could prevent the microactuator from damaging some biological samples (*e.g.*, cells and bacteria) during operation. In addition, we would to express that compared to the traditional ways of doping light-absorbing nanoparticles or metal layer deposition, the MJMA fabricated based on a two-in-one femtosecond (Fs) laser printing strategy requires only **10 mW** of actuation power and **30 ms** of response time, which is comparable to the best-performing light-triggered microactuators (Table R3.1)^{1~4}. Finally, we recognize the need to highlight the other two significant contributions of our research., including the **two-in-one processing strategy** of hydrogels and silver nanoparticles (Ag NPs), **multiple deformation modalities (>10)** achieved by multiple microjoints design and their independent deformation control, as described below.

Soft actuators	*Modality numbers	Size	Structure	Response time	Response Materials	Light stimuli mode
Ref. ¹	2	>1 cm	2D	>1 s	Hydrogel	Light (175~900 mW)
Ref. ²	2	<100 pm	2D	<100 ms	Hydrogel	Light (2.5 W)
Ref. ³	>10	>200 pm	2D	NA	Liquid crystal elastomer	2D structured light (2.5 W)
Ref. ⁴	>10	>500 pm	2D to 3D	NA	Shape memory alloy	2D scanning light (188 mW)
This work	>10	<100 pm	3D	30 ms	Hydrogel	3D structured light (<10 mW)

Table R3.1 Comparison of current light-triggered actuators.

*Modalities numbers: Here we only consider the number of modalities in the steady state.

References

1. Zhao, Y. et al. Soft phototactic swimmer based on self-sustained hydrogel oscillator. *Sci. Robot.* **4**, eaax7112 (2019).
2. Zhang, H., Mourran, A. & Moller, M. Dynamic switching of helical microgel ribbons. *Nano Lett.* **17**, 2010-2014 (2017).
3. Palagi, S. et al. Structured light enables biomimetic swimming and versatile locomotion of photoresponsive soft microrobots. *Nat. Mater.* **15**, 647-653 (2016).
4. Han, M. et al. Submillimeter-scale multimaterial terrestrial robots. *Sci. Robot.* **7**, eabn0602 (2022).

There is not much novelty in materials, fabrication process or structural design, thus not commensurate with the level of other papers published on Nature Comm.

Response: Based on your concerns, we have added the appropriate comparison, discussions, and revisions to the revised manuscript. We hope this will address your concerns about the novelty and impact of our work.

With the development of micro and nanofabrication technologies, researchers have shown great interest in soft microactuators. These microactuators primarily undergo deformation through environmental stimulus responses, such as magnetic fields, solvents, pH, temperature, and light. However, existing processing techniques have faced challenges in achieving a controlled distribution of stimulus-responsive units, thereby limiting the degree of deformation freedom. Moreover, triggering the local region deformation of microactuators in 3D space remains challenging. In our work, we are committed to solving the current limitations of limited deformation degrees and unprecise local region deformation control of microactuators. **Inspired by human joints, we design and built multiple microjoints** in the microactuator based on a **two-in-one Fs laser printing strategy** to extend the deformation degrees of freedom. In addition, through precise light distribution control in 3D space, MJMAs could exhibit **multiple deformation modalities (>10)**, thereby completing multiple micro cargos collection comparable to a robotic arm in a factory. In conclusion, compared with the state-of-the-art works, this paper mainly focused on the following two crucial novelties.

One (**two-in-one processing strategy**) is the combination of hydrogel and metal NPs at the microscale based on the same Fs laser system to extend the degree of the microactuator's deformation freedom. In previous studies, metal materials with light absorbing capability are usually added to the polymer surface or homogeneously within the polymer by physical vapor deposition² or doping⁵. However, both approaches introduce two main limitations, 1) doping often requires complex modifications to the surface of the microparticles to allow for uniform dispersion in the hydrogel precursor, which poses a high technical threshold for non-chemistry researchers. 2) Doping and deposition impart metal nanoparticles uniformly throughout the hydrogel so that the entire microstructure deforms in response to light-triggering. The structure is therefore difficult to accomplish local region response deformation (especially at the microscale), which limits the actuating freedom degree of the microactuator. Based on our proposed two-in-one Fs laser processing strategy, we can overcome these two limitations. Firstly, all commercial chemical reagents do not undergo additional operations before use and can be used only by simple mixing and stirring, which will not cause barriers to non-chemical researchers. In addition, we can precisely deposit Ag NPs on the specific location of hydrogel, where the Ag NPs and hydrogel together form a joint that can respond to light-triggering. Thus, our approach enables the designable spatial distribution of photothermal response units (joints) on the microactuator, thereby extending its degrees of freedom for deformation. Meanwhile, microactuators demonstrate deformation at **low actuation power (10 mW)** and **short response time (30 ms)**, which is comparable to the best performance of the reported microactuator.

Another one (**multiple deformation modalities**) is the realization of multiple deformation modalities of microactuators in 3D space. This achievement is made possible through the design of multiple micro-joints and their independent light-triggering control. Despite the significant advancements in stimuli-responsive actuators, their remote triggering control is often imprecise, and they offer only a limited number of deformation modalities. Typically, existing microactuators can only switch between two modalities using global stimuli such as pH, heat, or magnetic fields due to the lack of specific structural design and localized region stimulation. To extend the deformation modalities of microactuators, multiple micro joints are integrated into the microactuators inspired by human joints. Here, one micro joint is equivalent to one degree of deformation freedom. Therefore, the design of multiple micro-joints increases the degrees of deformation freedom of the microactuator, which will

greatly increase its modality numbers. In addition, the independent control of each joint and the multi-joint linkage are also key to the realization of microactuators with multiple deformation modalities. To date, scanning light³, 2D structured light⁴, and multi-wavelength beams⁶ approaches have been proposed to expand the modalities of the microactuator. However, the above methods still do not enable precise stimulation of localized regions of the microactuator in 3D space. In our work, based on the dynamic 3D multi-foci light, we can precisely stimulate each joint in 3D space, thereby linking the joints to achieve a variety of humanoid modalities (>10), such as raising hands, legs, and feet. More importantly, as a proof of concept, the micro robotic arm could collect micro cargos at different locations through the cooperation of multiple joints, which is similar to the robot in the factory.

In summary, we believe the MJMAs **hit the important challenges** in the field of soft microrobotics, regarding **miniaturized size, degree of actuation freedom, and precise triggering region control** (Just as reviewer #1 said), which can bring many new opportunities for soft microrobotics.

In response to your comments, we have emphasized the novelty and impact of this work in the revised manuscript. To provide the state-of-the-art of MJMAs, soft actuators have been comprehensively summarized in Supplementary Table 1. Compared to the existing soft actuators, our microactuators constructed by two-in-one Fs printing demonstrate multiple modality numbers (>10), much smaller 3D structure sizes (<100 μm), precise local region response (30 ms), and low actuation power (<10 mW). In addition, we have added seven figures (Supplementary Fig. 2, Fig. 3, Fig. 7, Fig. 8, Fig. 9, Fig. 13, and Fig. 16) in supplementary materials to further improve our manuscript. We hope that such revisions satisfactorily address your concerns and convince you that our work is suitable for publication in Nature Communications.

Reference

2. Zhang, H., Mourran, A. & Moller, M. Dynamic switching of helical microgel ribbons. *Nano Lett.* **17**, 2010-2014 (2017).
3. Palagi, S. et al. Structured light enables biomimetic swimming and versatile locomotion of photoresponsive soft microrobots. *Nat. Mater.* **15**, 647-653 (2016).
4. Han, M. et al. Submillimeter-scale multimaterial terrestrial robots. *Sci. Robot.* **7**, eabn0602 (2022).
5. Zheng, C.L. et al. Light-driven micron-scale 3D hydrogel actuator produced by two-photon

polymerization microfabrication. *Sens. Actuators B Chem.* **304**, 127345 (2020).

- Zuo, B., Wang, M., Lin, B.P. & Yang, H. Visible and infrared three-wavelength modulated multi-directional actuators. *Nat. Commun.* **10**, 4539 (2019).

List of Revisions:

V We revised the corresponding description on page 1 line 19: “Here, we propose a two-in-one femtosecond laser printing strategy to fabricate microjoints composed of hydrogel and metal nanoparticles, and develop multi-joint microactuators with multi-deformation modalities (>10), requiring short response time (30 ms) and low actuation power (<10 mW) to achieve deformation. Our microactuators with multiple modalities will bring many potential application opportunities in microcargo collection, microfluid operation, and cell manipulation.”

V We revised the corresponding description on page 4 line 67: “In this work, we propose a two-in-one femtosecond laser processing strategy to construct light-triggered multi-joint microactuators (MJMAs) with multiple deformation modalities (>10). The two-in-one processing strategy includes the construction of bio-inspired microjoints using asymmetric two-photon polymerization techniques⁵³⁻⁵⁶ and the deposition of silver nanoparticles (Ag NPs) onto hydrogel joints based on photoreduction techniques. Due to the strong photothermal conversion effect of the densely stacked Ag NPs, the microactuator exhibits short response time (30 ms) and low actuation laser power (<10 mW), even lower than the processing laser power (33 mW) of the microactuator. In addition, the independent deformation of each microjoint could be achieved by precise multi-foci light distribution control.”

V We added the corresponding description on page 11 line 227: “To provide the state-of-the-art of MJMAs, soft actuators have been comprehensively summarized in Supplementary Table 1. Compared to the existing soft actuators, our microactuators constructed by two-in-one Fs printing demonstrate multiple modality numbers (>10), much smaller 3D structure sizes (<100 μm), precise local region response (30 ms), and low actuation power (<10 mW).”

V We revised the corresponding description on page 14 line 290: “Although many efforts have been devoted to developing light-triggered soft actuators^{42, 68-70}, the open challenges of limited 2D morphologies, slow response speed, and inaccurate light-triggering have restricted their further innovation. The MJMAs in the current work demonstrate multi-deformation modalities (>10), low actuation power (<10 mW), and short response time (30 ms) in 3D space and manipulate multiple

microcargos at different locations.

In summary, we propose a two-in-one processing strategy to construct light-triggered MJMAs, which exhibit multiple deformation modalities (>10) in 3D space. The strong photothermal conversion effect of densely stacked Ag NPs results in low actuation power (10 mW) and short response time (30 ms) of the MJMAs. In addition, we not only realize the independent deformation control of each joint of MJMAs, but also can complete multi-joint linkage control in 3D space, thus making MJMAs exhibit multiple deformation modalities, including raising hand, leg, and foot. Ultimately, as a proof-of-concept, the double-joint micro robotic arm can collect microspheres at different locations.”

Below are some specific comments:

1. It is not clear if the AgNPs are deposited in the hydrogel or on the surface of the hydrogel.

Response: Thank you for the reviewer’s comment. To illustrate the position relationship between Ag NPs and hydrogels, we supplemented the relevant experiments. The laser-reduced nanoparticles (marked in yellow) can be seen stacked in the central region of the microplate in the top view of the electron microscope. To further verify the relationship between nanoparticles and hydrogels, we cut 2 μm with a focused ion beam (FIB) in the direction of the depth of Ag NPs. The side view SEM of the microstructure shows that the Ag NPs are mainly deposited on the surface of the hydrogel to achieve light-triggered actuation (Fig. R3.1).

Fig. R3.1 After focused ion beam (FIB) cutting, Ag NPs (yellow) can be seen deposited on the surface of the hydrogel in the side view (48°).

List of Revisions:

□ We added the corresponding description on page 7 line 94: “*In the second step, silver nanoparticles (Ag NPs) are selectively deposited on the surface of the humanoid robot joint by light reduction (Fig. 1d and Supplementary Fig. 2).*”

□ We added the Supplementary Fig. 2 and the corresponding references in the supplementary materials.

2. It is not clear whether the motion of the microactuator is triggered in water or in air.

Response: We apologize for bringing the misunderstanding to the reviewer. In our work, since the main body of the microactuator is composed of thermal responsive hydrogel, its deformation principle is a reversible phase transition from a hydrophilic expansion state below its LCST to a hydrophobic collapse state above its LCST⁷. Therefore, all demonstrations are done in a liquid environment. Reference:

7. Tang, L. et al. Poly(N-isopropylacrylamide)-based smart hydrogels: Design, properties and applications. *Prog. Mater. Sci.* **115**, 100702 (2021).

List of Revisions:

□ We added the corresponding description on page 16 line 342: “*All of the light-triggered microactuators were actuated in deionized water.*”

3. Line 213, the subtitle should be “deformation in 3D space”.

Response: We apologize for this grammar mistake. And we corrected it in our revised manuscript.

List of Revisions:

□ We revised the corresponding description on page 11 line 232: “*Multi-joint linkage deformation in 3D space*”

REVIEWERS' COMMENTS

Reviewer #1 (Remarks to the Author):

The authors have satisfactorily addressed my comments, and I recommend their work for publication. Here are a few points regarding the comments raised by the Reviewer #2, #3.

#2, In my opinion, rheology and viscosity measurements are irrelevant because the fabrication technique relies on a two-photon laser writer rather than an injection-based 3D printer.

#3, I believe I have a comprehensive understanding of the viewpoints expressed by both sides, namely the reviewer's opinion that there is "not much novelty in materials, fabrication process, or structural design," and the authors' assertion of "novelties in two-in-one processing and multiple deformation modalities." However, I do hold some slightly divergent perspectives:

I have been contemplating for a considerable amount of time about the possible techniques to integrate light-sensitive components (in a selective and localized manner) onto a microstructure for microactuation. The addition of pigments into the material is not effective, as they tend to get washed away easily or interfere with the polymerization process by reducing the two-photon absorption of the femtosecond (fs) beam. Therefore, I believe that the authors' employment of laser deposition of silver particles is an astute and efficient solution.

I still believe that using the phrase "Inspired by human joints..." is excessive. Bioinspiration may not be an appealing selling point to enhance the uniqueness of the manuscript. However, I am inclined to view this matter as a difference in aesthetic preference among individuals. Therefore, this comment does not impact the scientific aspects of the manuscript.

For the potentials I see, (1) by using SLM, the hydrogel structure can achieve highly sophisticated soft robotic manipulation at the truly microscopic scale. The structures are highly deformable and surpass the capability of multiple-beam optical tweezers. (2) The gel could also utilize locomotion strategies that exist in nature, such as those employed by sperm and protozoa. The authors may consider adding a few sentences to the discussion, although this is ultimately up to them. The decision-making process lies with the editor, while we (reviewers) provide only scientific comments based on their background, which may introduce some bias (my comments).

Hao Zeng

Reviewer #2 (Remarks to the Author):

The authors have carefully addressed all my concerns in details. Having carefully gone through the 1st and 3rd reviewers' comments. After carefully going through the revised manuscript and the responses to reviewers, it seems authors have addressed the issues raised. Overall, I think that the revised manuscript has its own unique qualities and originality. I believe that the publication of this interesting work would have a wide range of audience. Therefore, I recommend this manuscript to be directly accepted for publication in Nature Communication as its current form.

Reviewer #3 (Remarks to the Author):

The authors clearly addressed the novelties and the values of the work in the response letter and revised manuscript. The two-in-one processing and multiple deformation modalities will bring good impacts. Therefore, I recommend this manuscript to be published on Nature Communications.

Reviewer #1 (Remarks to the Author):

The authors have satisfactorily addressed my comments, and I recommend their work for publication. Here are a few points regarding the comments raised by the Reviewer #2, #3.

#2, In my opinion, rheology and viscosity measurements are irrelevant because the fabrication technique relies on a two-photon laser writer rather than an injection-based 3D printer.

#3, I believe I have a comprehensive understanding of the viewpoints expressed by both sides, namely the reviewer's opinion that there is "not much novelty in materials, fabrication process, or structural design," and the authors' assertion of "novelties in two-in-one processing and multiple deformation modalities." However, I do hold some slightly divergent perspectives:

I have been contemplating for a considerable amount of time about the possible techniques to integrate light-sensitive components (in a selective and localized manner) onto a microstructure for microactuation. The addition of pigments into the material is not effective, as they tend to get washed away easily or interfere with the polymerization process by reducing the two-photon absorption of the femtosecond (fs) beam. Therefore, I believe that the authors' employment of laser deposition of silver particles is an astute and efficient solution.

Response: We thank the referee very much for the review and high recognition of our work. We also thank the referee for recommending our work for publication in Nature Communications.

I still believe that using the phrase "Inspired by human joints..." is excessive. Bioinspiration may not be an appealing selling point to enhance the uniqueness of the manuscript. However, I am inclined to view this matter as a difference in aesthetic preference among individuals. Therefore, this comment does not impact the scientific aspects of the manuscript.

Response: We thank the reviewer for the open attitude in terms of personal aesthetic preference. We believe that there is a good similarity between micro joints and human joints and therefore retain the relevant descriptions.

For the potentials I see, (1) by using SLM, the hydrogel structure can achieve highly sophisticated soft robotic manipulation at the truly microscopic scale. The structures are highly deformable and surpass the capability of multiple-beam optical tweezers. (2) The gel could also utilize locomotion strategies that exist in nature, such as those employed by sperm and protozoa. The authors may consider adding a few sentences to the discussion, although this is ultimately up to them. The decision-making process lies with the editor, while we (reviewers) provide only scientific comments based on their background, which may introduce some bias (my comments).

Hao Zeng

Response: Many thanks for the reviewer's kind advice. According to the reviewer's suggestion, we have added some appropriate discussion of MJMA's advantages and prospects in the manuscript to further improve the quality of the article.

List of Revisions:

- We added the discussion on page 14 line 296: *“In particular, by enabling multi-foci beams through SLM, hydrogel microstructures allow for highly complex soft robotics manipulations at the micro-scale. These structures demonstrate strong deformable capabilities, exceeding the ability of multi-beam optical tweezers to manipulate microcargos. In addition, hydrogel microactuators could also be adapted to the needs of manipulation in dynamic environments by taking advantage of the locomotion strategies that exist in nature, such as sperm and protozoa. For example, we could introduce magnetic materials into existing microactuators to build movable microrobot systems with multiple programmable modalities, that is, magnetic fields control locomotion, and light fields control deformation.”* was added.
- To further highlight the innovative nature of the processing approach in this study, we revised the title to *“Light-triggered multi-joint microactuator fabricated by two-in-one femtosecond laser writing”*

Reviewer #2 (Remarks to the Author):

The authors have carefully addressed all my concerns in details. Having carefully gone through the 1st and 3rd reviewers' comments. After carefully going through the revised manuscript and the responses to reviewers, it seems authors have addressed the issues raised. Overall, I think that the revised manuscript has its own unique qualities and originality. I believe that the publication of this interesting work would have a wide range of audience. Therefore, I recommend this manuscript to be directly accepted for publication in Nature Communication as its current form.

Response: We sincerely thank the reviewer for the kind comments and helping us improve the quality.

Reviewer #3 (Remarks to the Author):

The authors clearly addressed the novelties and the values of the work in the response letter and revised manuscript. The two-in-one processing and multiple deformation modalities will bring good impacts. Therefore, I recommend this manuscript to be published on Nature Communications.

Response: Many thanks to the reviewer for careful review and valuable comments of this work. We also thank the referee for recommending our work for publication in Nature Communications.